

# Algebraic law of local correlations in a driven Rydberg atomic system

Xin Wang[1], XiaoFeng Wu[1], Bo Yang[2], Bo Zhang[1] and Bo Xiong[1]⋆

**1** Department of Physics, Wuhan University of Technology, Wuhan 430070, China
**2** School of Mathematics and Physics, Hubei Polytechnic University, Huangshi 435003, China

⋆ boxiong@whut.edu.cn

## Abstract

Understanding the mechanism behind the buildup of inner correlations is crucial for studying nonequilibrium dynamics in complex, strongly interacting many-body systems. Here we investigate both analytically and numerically the buildup of antiferromagnetic (AF) correlations in a dynamically tuned Ising model with various geometries, realized in a Rydberg atomic system. Through second-order Magnus expansion (ME), we demonstrate quantitative agreement with numerical simulations for diverse configurations including $2 \times n$ lattice and cyclic lattice with a star. We find that the AF correlation magnitude at fixed Manhattan distance obeys a universal superposition principle: It corresponds to the algebraic sum of contributions from all shortest paths. This superposition law remains robust against variations in path equivalence, lattice geometries, and quench protocols, establishing a new paradigm for correlation propagation in quantum simulators.

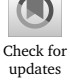

## 1   Introduction

Understanding nonequilibrium dynamics of complex, strongly interacting many-body systems is an important and challenging issue at the intersection between statistical physics and quantum physics [1, 2]. A crucial goal in studying such dynamics is to explore the buildup of correlations and their time evolution, especially near quantum phase transitions [3]. Previous theoretical works have established that the dynamical buildup of quantum correlations — manifested as topological defect formation [4–7] or squeezing of quantum fluctuations [8] — universally emerges when systems are driven through critical points at finite rates. This is rigorously demonstrated via exact solutions [7], mean-field expansions [8], and generalized frameworks for tunable transitions [5], revealing universal scaling laws governed by critical slowing down. However, while these advances provide a comprehensive framework for uniform systems, they face fundamental limitations in addressing nonuniform quantum many-body dynamics. The exponential growth of the Hilbert space prevents exact solutions [7] and overwhelms numerical simulations, while the spatially inhomogeneous interactions render mean-field expansions [8] intractable. Consequently, developing an efficient theory for *nonuniform* nonequilibrium dynamics becomes imperative to decode: (i) the spatially resolved buildup of correlations [9, 10], (ii) emergent critical dynamics [11–13], and (iii) quantum resource generation [14, 15] in next-generation simulations.

Recent advances in Rydberg atom platforms have enabled the realization of programmable lattice arrays with exotic geometries, including triangular [10, 15], Kagome [16, 17], and honeycomb [18] lattices, as well as finite-configuration prototypes such as star, cyclic, diamond, and hexagon structures [19]. The precise control over interaction strengths and Rabi frequencies in these systems [14, 20] has facilitated experimental investigations of fundamental questions in many-body nonequilibrium dynamics, such as quantum Kibble-Zurek mechanism [21, 22], quantum scar states [18], and quantum phase transitions [10, 13]. Furthermore, the versatility of geometric configurations in these systems establishes them as ideal quantum simulators for studying geometric effects on nonequilibrium dynamics in strongly correlated quantum matter. Here, we employ such geometrically diverse Rydberg arrays to study quench dynamics in tunable Ising models with spatial inhomogeneity. Through systematic ME analysis, we derive closed-form expressions for connected correlation functions that explicitly reveal the AF correlation buildup mechanism. Our key discovery identifies a universal path superposition principle: The AF correlation magnitude at fixed Manhattan distance is governed by the algebraic sum of contributions from all shortest paths. This law is independent of path equivalence, lattice geometries, and quench protocols, implying that complex correlation networks can be deconstructed into elementary path contributions. Full quantum simulations using time-dependent Schrödinger equation for scalable systems confirm the generality and robustness of this principle.

Figure 1: (a) Schematic description of the lattice, and atoms are excited from the ground state $|g\rangle$ to the Rydberg state $|r\rangle$ through a two-photon process. (b) The protocol of Rabi frequency $\Omega(t)$ and detunning $\delta(t)$.

## 2 Model and method

### 2.1 The effective two-level Ising-like Rydberg system

Our model is based on recent experiments with $^{87}$Rb atoms [10], in which the atoms are manipulated in user-defined two-dimensional optical tweezer arrays – each containing a single atom – to probe nonequilibrium dynamics. In the experimental sequence, atoms are first prepared in the hyperfine ground state $|g\rangle = |5S_{1/2}, F = 2, m_F = 2\rangle$ via optical pumping. Subsequently, a two-photon transition through the intermediate state $|e\rangle = |5P_{1/2}, F = 2\rangle$ coherently couples $|g\rangle$ to the Rydberg state $|r\rangle = |64D_{3/2}, m_j = 3/2\rangle$, with counter-propagating 795 nm and 475 nm lasers driving the $|g\rangle \rightarrow |e\rangle$ and $|e\rangle \rightarrow |r\rangle$ transitions respectively. The experimental configuration thus realizes a three-level system comprising states $|g\rangle$, $|e\rangle$, and $|r\rangle$. The $|g\rangle \leftrightarrow |e\rangle$ transition is driven with Rabi frequency $\Omega_e$ and detuning $\delta_e$, while the $|e\rangle \leftrightarrow |r\rangle$ transition has Rabi frequency $\Omega_r$ and detuning $\delta_r$ [see Fig. 1 (a)]. Experiments usually modulate these parameters as $|\delta_e| \approx |\delta_r| \gg |\delta_t| \equiv |\delta_e + \delta_r|$ to establish a two-photon resonance condition between $|g\rangle$ and $|r\rangle$: (a) The near-cancellation $\delta_e \approx -\delta_r$ minimizes the two-photon detuning $\delta_t$, enabling resonant coupling between $|g\rangle$ and $|r\rangle$ through virtual transitions via $|e\rangle$. (b) The large single-photon detunings ($|\delta_e|, |\delta_r| \gg |\delta_t|$) suppress direct population transfer to $|e\rangle$, effectively decoupling it from the dynamics and creating an effective two-level system between $|g\rangle$ and $|r\rangle$. The Hamiltonian governing the interaction between a three-level atom and laser fields is expressed as (with $\hbar = 1$):

$$H_1^{'} = \omega_e |e\rangle\langle e| + \omega_r |r\rangle\langle r| + \frac{\Omega_e}{2}\left(e^{-i\omega_1 t}|e\rangle\langle g| + \text{h.c.}\right) + \frac{\Omega_r}{2}\left(e^{-i\omega_2 t}|r\rangle\langle e| + \text{h.c.}\right),$$

where the ground state energy is set as the zero reference point. We transform to a rotating frame through the unitary operator $U = |g\rangle\langle g| + e^{i\omega_1 t}|e\rangle\langle e| + e^{i(\omega_1+\omega_2)t}|r\rangle\langle r|$. The transformed state in this frame is given by $|\psi\rangle = U|\psi^{'}\rangle$, where $|\psi^{'}\rangle$ denotes the three-level state. Substituting into the time-dependent Schrödinger equation gives $H_1 = UH_1^{'}U^\dagger - iU\frac{dU^\dagger}{dt}$. Explicit calculation yields the Hamiltonian:

$$H_1 = -\delta_t|r\rangle\langle r| - \delta_e|e\rangle\langle e| + \frac{\Omega_e}{2}(|e\rangle\langle g| + \text{h.c.}) + \frac{\Omega_r}{2}(|r\rangle\langle e| + \text{h.c.}),$$

where the detunings are defined as $\delta_e = \omega_1 - \omega_e$ and $\delta_t = \omega_1 + \omega_2 - \omega_r$. Here, non-resonant terms have been neglected under the rotating wave approximation, preserving only resonant photon exchange processes. For constructing an effective two-level Hamiltonian, we partition $H_1$ into diagonal ($H_P$, $H_Q$) and coupling ($T_S$, $T_S^\dagger$) terms:

$$H_P = -\delta_t|r\rangle\langle r|, \qquad\qquad H_Q = -\delta_e|e\rangle\langle e|,$$

$$T_S = \frac{\Omega_e}{2}|g\rangle\langle e| + \frac{\Omega_r}{2}|r\rangle\langle e|, \qquad\qquad T_S^\dagger = \frac{\Omega_e}{2}|e\rangle\langle g| + \frac{\Omega_r}{2}|e\rangle\langle r|.$$

Partitioning the Hilbert space into $P = \{|g\rangle, |r\rangle\}$ and $Q = \{|e\rangle\}$ subspaces, the Schrödinger equation becomes:

$$\begin{pmatrix} H_P & T_S \\ T_S^\dagger & H_Q \end{pmatrix} \begin{pmatrix} \psi_P \\ \psi_Q \end{pmatrix} = E \begin{pmatrix} \psi_P \\ \psi_Q \end{pmatrix}.$$

Next, we obtain $\psi_Q = \frac{1}{E - H_Q} H_S^\dagger \psi_P$ from the $Q$-block equation and substitute it into $P$-block so that we acquire

$$H_{\text{eff}} = H_P + T_S \frac{1}{E - H_Q} T_S^\dagger. \tag{1}$$

Through the adiabatic elimination ($|E| \ll |\delta_e|$), and substituting $H_P$, $H_Q$, $T_S$, and $T_S^\dagger$ into Eq. (1), we obtain the effective Hamiltonian for $P$ subspace,

$$H_{\text{eff}} = \frac{\Omega_e^2}{4\delta_e} |g\rangle\langle g| - \left( \delta_t - \frac{\Omega_r^2}{4\delta_e} \right) |r\rangle\langle r| + \frac{\Omega_e \Omega_r}{4\delta_e} \left( |g\rangle\langle r| + |r\rangle\langle g| \right).$$

Setting the ground state energy to zero,

$$H_{\text{eff}} = -\left( \delta_t + \frac{\Omega_e^2 - \Omega_r^2}{4\delta_e} \right) |r\rangle\langle r| + \frac{\Omega_e \Omega_r}{4\delta_e} \left( |g\rangle\langle r| + |r\rangle\langle g| \right),$$

where the effective detuning is $\delta = \delta_t + \frac{\Omega_e^2 - \Omega_r^2}{4\delta_e}$, and the effective Rabi frequency is $\Omega = \frac{\Omega_e \Omega_r}{2\delta_e}$. Note that the two lasers generate an effective Rabi frequency $\Omega$ for the coupling between $|g\rangle$ and $|r\rangle$, while also introducing lightshifts $\frac{\Omega_e^2 - \Omega_r^2}{4\delta_e}$, which contribute to the two-photon detuning $\delta$. To achieve independent control of $\Omega$ and $\delta$, the experiments compensate the lightshifts via real-time feedforward correction to the laser frequencies and tune $\Omega$ dynamically using acousto-optic modulators.

Taking the interactions between atoms in $|r\rangle$ into account, the full Hamiltonian for this system can be written as,

$$H = \frac{\hbar\Omega(t)}{2} \sum_i \sigma_i^x - \hbar\delta(t) \sum_i n_i + \sum_{\langle ij \rangle} U_{ij} n_i n_j, \tag{2}$$

where $\sigma_i^x = |g\rangle\langle r|_i + |r\rangle\langle g|_i$ is the $x$-Pauli matrix, representing the transition operator at site $i$ between $|g\rangle$ and $|r\rangle$. $n_i = |r\rangle\langle r|_i$ is the projection operator for the Rydberg state at site $i$, which is related to the $z$-Pauli matrix by $n_i = \frac{1}{2}(\sigma_i^z + 1)$. The interaction term $U_{ij} \sim 1/r_{ij}^6$ is the van der Waals interaction between the $i$th and $j$th Rydberg atoms, where $r_{ij}$ is the distance between them. Due to the strong, repulsive interaction, atoms within a critical distance cannot occupy $|r\rangle$ simultaneously; this is a so-called Rydberg blockade [23]. We consider the Rydberg blockade radius $R_b \cong r_{\langle ij \rangle}$, within which adjacent atoms are strongly prevented from being excited simultaneously to $|r\rangle$. The Hamiltonian (2) is regarded as the quantum Ising model where a transverse field $B_\perp \propto \Omega$, a longitudinal field $B_\parallel \propto \delta$ and Ising coupling $U_{ij}$. The associated equilibrium state for $U > 0$ displays two phases, paramagnetic (PM) and AF phases, with a second-order phase transition between them.[1] In magnetic systems, the PM phase corresponds to a disordered spin arrangement with no long-range order, while the AF phase is characterized by neighboring spins aligning in opposite directions.

---

[1]Although the term $\delta$ favors breaking the $\mathbb{Z}_2$ symmetry (which typically suppresses second-order phase transitions driven by spontaneous symmetry breaking), the strong suppression of $\delta$ by $U_{ij}$, combined with the fact that $B_\parallel$ is negligible compared to $\Omega$ and $U_{ij}$, allows the system to exhibit a second-order phase transition.

Table 1: Parameters used for the numerical and analytic results presented in the main text (all parameters are given in MHz).

| Figures | Structures | $U_1/h$ | $U_2/h$ | $U_3/h$ | $\Omega_{\mathrm{max}}/(2\pi)$ | $\delta_f/(2\pi)$ |
|---|---|---|---|---|---|---|
| 3 | $2 \times n$ lattice | 2.8 | 1.4 | / | 1.8 | $[1, 5]$ |
| 4, 6, 7 | cyclic lattice with a star | 2.8 | 1.4 | 3.0 | 0.8 | $[1, 5]$ |
| 5 (a) | hexagonal lattice | 3.0 | 2.0 | 2.5 | 1.8 | $[1, 5]$ |
| 5 (b) | octagonal lattice | | | | | |

## 2.2 Quench protocol and antiferromagnetic correlation

To explore the buildup of AF correlations, we implement a quench protocol following the design of recent experiments [10], as shown in Fig. 1 (b). The protocol initiates with all atoms prepared in the $|g\rangle$ state. During the first stage, we ramp up the Rabi frequency $\Omega$ from zero to $\Omega_{\mathrm{max}}$ over a duration $t_{\mathrm{rise}} = 0.1\mu s$, maintaining a constant negative detuning $\delta = \delta_0$. Subsequently, we hold $\Omega$ at $\Omega_{\mathrm{max}}$ for a time interval $t_{\mathrm{sweep}} = 0.5\mu s$ while linearly sweeping the detuning from $\delta_0$ to a final positive value $\delta_f$, thereby driving the system into the AF phase. Finally, we ramp down $\Omega$ back to zero over a time $t_{\mathrm{fall}} = 0.1\mu s$ while keeping $\delta$ fixed at $\delta_f$. The emergence of AF ordering throughout this sequence can be quantified through the connected spin-spin correlation

$$C_{kl} = \frac{1}{N_{kl}} \sum_{(ij)} \left[ \langle n_i n_j \rangle - \langle n_i \rangle \langle n_j \rangle \right], \tag{3}$$

where $\langle n_i \rangle$ denotes the expectation value of the occupation number at site $i$, and the summation runs over all atom pairs with relative displacement $(ka, lb)$ in the 2D lattice. Here $a$ and $b$ are the lattice constants along orthogonal axes, and $N_{kl}$ counts the number of valid pairs for each displacement sector. The anisotropic interaction profile required for this measurement can be engineered through controlled positioning of Rydberg atoms using optical tweezers combined with dynamic regulation of external electric fields that modify the interaction tensor components [24, 25].

We perform numerical simulations using the fourth-order Runge-Kutta method [26] for various lattice geometries with periodic boundary conditions along the row. These geometries include the $2 \times n$ lattice, the cyclic lattice with a star, the hexagonal lattice and the octagonal lattice. Within the considered time scale, the numerical calculations reveal similar results for the connected correlation functions across different lattice lengths, e.g., $2 \times 8$, $2 \times 10$, and $2 \times 12$ in the $2 \times n$ lattice. The characteristic parameters in our simulations – including the maximum Rabi frequency $\Omega_{\mathrm{max}}/(2\pi) \approx 1.8\,\mathrm{MHz}$, final detuning $\delta_f/(2\pi) \in [1,5]\,\mathrm{MHz}$, and interaction strengths $U_1/h = 2.8\,\mathrm{MHz}$ along principal axes – are directly adopted from the Rydberg quantum simulator implementation in Ref. [10], with complete numerical values tabulated in Table 1. The analytic results for different lattice geometries are shown in Appendix A.

## 2.3 Magnus expansion

The nonequilibrium dynamics of the Ising-like models poses significant theoretical challenges for analytical studies due to the exponential growth of the Hilbert space dimension with increasing system size. However, recent advances [27] demonstrate that exact analytical solutions can be constructed for systems with local interactions and finite on-site Hilbert spaces on $n$-dimensional hypercubic lattices. In this work, we focus on the finite-time regime, where quantum coherence dominates before the system evolves into chaotic or thermalized states.

Within this regime, we employ ME to derive analytical expressions for key dynamical observables, particularly the connected spin-spin correlation function. These analytical results are then compared with exact numerical solutions. ME is a practical way to build up approximate exponential representations of the solution of linear systems of differential equations with varying coefficients [28]. It has been applied widely in some areas, from atomic and molecular physics [29,30] to nuclear magnetic resonance [31] to quantum electrodynamics and elementary particle physics [32,33].

Under the condition $\int_0^T \|H(t)\|_2 \, dt < \pi$, where $\|H\|_2$ is the euclidean/spectral norm of $H$ defined as the squared root of the largest eigenvalue of positive semi-definite operator $H^\dagger H$, the ME treatment of the many-body propagator yields the time-evolution operator $\hat{U}(T) = \exp\left[-iT\bar{H}(T)\right]$, where $\bar{H}(T)$ is a series expansion involving nested commutators of the time-dependent Hamiltonian (2), i.e., $\bar{H}(T) = \sum_{k=1}^{\infty} \bar{H}_k(T)$. To ensure computational tractability, we truncate the ME at second order: $\bar{H}_1 = \frac{1}{T} \int_0^T H(t_1) \, dt_1$ and $\bar{H}_2 = \frac{i}{2T} \int_0^T \left[\int_0^{t_1} H(t_2) dt_2, H(t_1)\right] dt_1$, where the commutator $[\hat{A}, \hat{B}] = \hat{A}\hat{B} - \hat{B}\hat{A}$. Although the Magnus expansion is an approximate method, the quantitative agreement between our analytical results and numerical simulations in various geometries and quench protocols supports the validity of the second-order ME for capturing the essential physics of the AF correlation buildup under the parameters and time scales considered. Furthermore, the rigorous proof of the self-consistency of ME, e.g., for $U = 0$, $C_{kl} = 0$, is shown in Appendix B. One may note that there exists an internal connection between Magnus series and Dyson perturbative series. The former derives the integral of the nested commutators due to the nonzero commutation of the Hamiltonians at different times and the latter causes the path integral raised by the time-order products .

For experimentally relevant detuning modulation, we adopt a linear ramp $\delta(t) = \frac{\delta_f - \delta_0}{T} t + \delta_0$, where $\delta_0$ is the original value and $\delta_f$ is the final value of the detuning. One can deduce straightforwardly ($\hbar = 1$), $\bar{H}_1 = \frac{\Omega}{2} \sum_i \sigma_i^x - \delta_{\text{avg}} \sum_i n_i + \sum_{\langle ij \rangle} U_{ij} n_i n_j$ and $\bar{H}_2 = \frac{\Omega}{24}(\delta_f - \delta_0)T \sum_i \sigma_i^y$ with $\delta_{\text{avg}} = \frac{\delta_0 + \delta_f}{2}$. Note that the final term in $\bar{H}$ is linearly dependent on $T$, which is originated from the second-order ME. We can expand the matrix exponential into $\hat{U}(T) = \sum_{n=1}^{\infty} \frac{(-iT)^n}{n!} \bar{H}^n(T)$, where the high-power terms in $T$ appear and will contribute in the correlation function .

To investigate the impact of paths between the grid point $(0,0)$ and $(ka, lb)$ on the buildup of AF correlations, we analyze the quantity $C_R(T) = C_{((0,0),(k,l))}$, where the Manhattan distance is defined as $R = |k| + |l|$. In the Magnus approach to the Ising-like Hamiltonian, the buildup of AF correlations, characterised by $C_R(T)$, requires expanding the exponential matrix $\hat{U}(T)$ to sufficiently high orders, where the interaction term becomes significant in the dynamic process. We emphasize that the non-zero commutation of the Hamiltonian at different time in $\bar{H}_2$ plays a crucial role in the ME, and thus it must be taken into account to obtain more accurate results.

## 3 Results

### 3.1 Nonequivalent paths

In this section, we first derive universal analytic expressions for the connected correlation function $C_R$ by systematically analyzing nonuniform lattice systems, explicitly identifying the physical origin of each term in the expressions. Furthermore, through applying the ME to a prototypical lattice geometry, we analytically demonstrate that $C_R$ emerges as the algebraic sum of contributions from all shortest paths between lattice sites. Crucially, these results – unob-

tainable within the uniform system framework of our prior work [34] – highlight the necessity of extending ME to nonuniform systems. Building upon our recent work that applied the ME to analyze nonequilibrium dynamics in uniform lattice systems, we extend this framework to nonuniform lattice systems.Through extensive analytic calculations across distinct lattice geometries (as detailed in Appendix A), we systematically derive universal analytic expressions for the AF correlation function $C_R$, explicitly obtaining results for the nearest-neighbor sites $(R = 1)$

$$C_{R=1} \qquad C_{R=1}^{\text{Path}} \qquad C_{R=1}^{\text{Coupling}} \qquad C_{R=1}^{\text{Time}}$$

and the next-nearest-neighbor sites $(R = 2)$

$$C_{R=2} \qquad C_{R=2}^{\text{Path}} \qquad C_{R=2}^{\text{Coupling}} \qquad C_{R=2}^{\text{Time}}$$

where the solid lines between two reference balls label the shortest paths and the wave lines denote the coupling from the nearest-neighbor sites to the shortest paths. To elucidate the physical origin of distinct contributions in the correlation function, we analyze $C_{R=1}$ through the ME framework. The correlation is decomposed into three components: (a) Path contribution ($C_{R=1}^{\text{Path}}$): Dominated by the leading-order ME term, this corresponds to the shortest-path propagation between adjacent lattice sites. (b) Coupling contribution ($C_{R=1}^{\text{Coupling}}$): Arising from higher-power terms in the first-order ME, it quantifies environmental coupling effects – specially, perturbations induced by nearest-neighbor sites on the shortest paths. (c) Time-dependent contribution ($C_{R=1}^{\text{Time}}$): Generated by the second-order ME term, this captures non-commutative quantum dynamics – a direct consequence of the nonzero commutation relation between the Hamiltonian at different times, i.e., $[H(t_1), H(t_2)] \neq 0$ for $t_1 \neq t_2$. Physically, this term encodes the temporal interference caused by the time-ordering of interactions, leading to corrections in the correlation buildup that cannot be ignored in nonequilibrium processes. Remarkably, the expressions are identical between any two reference points that share the same shortest paths in the short-time behavior of the AF correlation regardless of structural complexity, for example, $C_{R=1}^{\text{Path}}\left( \vphantom{\rule{0pt}{1em}} \right) = C_{R=1}^{\text{Path}}\left( \vphantom{\rule{0pt}{1em}} \right) = C_{R=1}^{\text{Path}}\left( \vphantom{\rule{0pt}{1em}} \right)$ [see Appendix A].

To investigate the impact of nonequivalent paths on the AF correlation $C_R$, we select the next-nearest-neighbor correlation $C_{11}$ in a square lattice as a prototypical example. Through ME calculations, we derive the analytic expression

$$C_{11} \qquad C_{11}^{\text{Path}} \qquad C_{11}^{\text{Coupling}} \qquad C_{11}^{\text{Time}}$$

(4)

where two red curves in $C_{11}$ denote two nonequivalent paths between the diagonal reference points. Notably, the analytic expressions exhibit a characteristic scaling structure in their denominators

$$\Omega^p \delta^q U^m T^n \, (p, q, m, n \in \mathbb{Z}; n = p + q + m),$$

where $\Omega$ and $T$ act as universal scaling factors across all terms, while $\delta$ and $U$ vary significantly between contributions. To simplify the representation, we define a dimensionless scaling function $\mathcal{M}_{(n,p)}$ by factoring out $\Omega$ and $T$. The exact forms of each contribution for $C_{11}$ yields:

$$C_{11}^{\text{Path}}(T) = \frac{T^{10}\Omega^6}{2419200}\mathcal{M}_{(10,6)}^{11}, \tag{5a}$$

$$\begin{aligned}
C_{11}^{\text{Coupling}}(T) = &-\frac{T^{12}\Omega^6}{58060800}\left[\mathcal{M}_{(12,6)}^{11} + \Omega^2\mathcal{M}_{(12,8)}^{11}\right] \\
&+ \frac{T^{14}\Omega^6}{53648179200}\left[\mathcal{M}_{(14,6)}^{11} + \Omega^2\mathcal{M}_{(14,8)}^{11} + \Omega^4\mathcal{M}_{(14,10)}^{11}\right],
\end{aligned} \tag{5b}$$

$$C_{11}^{\text{Time}}(T) = \frac{T^{12}\Omega^6}{116121600}\left[1 + \frac{T^2}{144}(\delta_f - \delta_0)^2 + \frac{T^4}{62208}(\delta_f - \delta_0)^4\right](\delta_f - \delta_0)^2\mathcal{M}_{(10,6)}^{11}. \tag{5c}$$

where

$$\mathcal{M}_{(10,6)}^{11} = \left[77(U_1^4 + U_2^4) - 340(U_1^3 + U_2^3)\delta_{\text{avg}} + 375(U_1^2 + U_2^2)\delta_{\text{avg}}^2\right],$$

$$\begin{aligned}
\mathcal{M}_{(12,6)}^{11} = \Big\{2\Big[&52(U_1^6 + U_2^6) - 368(U_1^5 + U_2^5)\delta_{\text{avg}} + 1037(U_1^4 + U_2^4)\delta_{\text{avg}}^2 \\
&- 1432(U_1^3 + U_2^3)\delta_{\text{avg}}^3 + 831(U_1^2 + U_2^2)\delta_{\text{avg}}^4\Big]\Big\},
\end{aligned}$$

$$\begin{aligned}
\mathcal{M}_{(12,8)}^{11} = \Big\{&593(U_1^4 + U_2^4) - 585(U_1^3 U_2 + U_1 U_2^3) - 448 U_1^2 U_2^2 \\
&- 88(U_1 + U_2)\left[28(U_1^2 + U_2^2) - 43 U_1 U_2\right]\delta_{\text{avg}} + 2544(U_1^2 + U_2^2)\delta_{\text{avg}}^2\Big\},
\end{aligned}$$

$$\begin{aligned}
\mathcal{M}_{(14,6)}^{11} = \Big\{24\Big[&121(U_1^8 + U_2^8) - 1170(U_1^7 + U_2^7)\delta_{\text{avg}} + 4952(U_1^6 + U_2^6)\delta_{\text{avg}}^2 \\
&- 11862(U_1^5 + U_2^5)\delta_{\text{avg}}^3 + 17112(U_1^4 + U_2^4)\delta_{\text{avg}}^4 - 14322(U_1^3 + U_2^3)\delta_{\text{avg}}^5 \\
&+ 5565(U_1^2 + U_2^2)\delta_{\text{avg}}^6\Big]\Big\},
\end{aligned}$$

$$\begin{aligned}
\mathcal{M}_{(14,8)}^{11} = \Big\{&11\left[2771(U_1^6 + U_2^6) - 4060(U_1^5 U_2 + U_1 U_2^5) - 7893(U_1^4 U_2^2 + U_1^2 U_2^4)\right] \\
&- 11436 U_1^3 U_2^3 - 6(U_1 + U_2)\left[34742(U_1^4 + U_2^4) - 76315(U_1^3 U_2 + U_1 U_2^3)\right. \\
&\left. + 514 U_1^2 U_2^2\right]\delta_{\text{avg}} + \left[577781(U_1^4 + U_2^4) - 538804(U_1^3 U_2 + U_1 U_2^3)\right. \\
&\left. - 762498 U_1^2 U_2^2\right]\delta_{\text{avg}}^2 - 84(U_1 + U_2)\left[9397(U_1^2 + U_2^2) - 14842 U_1 U_2\right]\delta_{\text{avg}}^3 \\
&+ 444780(U_1^2 + U_2^2)\delta_{\text{avg}}^4\Big\},
\end{aligned}$$

$$\begin{aligned}
\mathcal{M}_{(14,10)}^{11} = \Big\{&33\left[2438(U_1^4 + U_2^4) - 5971(U_1^3 U_2 + U_1 U_2^3) - 2926 U_1^2 U_2^2\right] \\
&- 21(U_1 + U_2)\left[15139(U_1^2 + U_2^2) - 34774 U_1 U_2\right]\delta_{\text{avg}} + 311220(U_1^2 + U_2^2)\delta_{\text{avg}}^2\Big\}.
\end{aligned}$$

To comparatively analyze the dual-path contributions in $C_{11}$ and the single-path configuration, we further calculate the corresponding single-path component $C_{20}$ for systematic evaluation:

$$
C_{20} \qquad C_{20}^{\text{Path}} \qquad C_{20}^{\text{Coupling}} \qquad C_{20}^{\text{Time}}
$$

$$\tag{6}$$

$$C_{20}^{\text{Path}}(T) = \frac{T^{10}\Omega^6}{2419200}\mathcal{M}_{(10,6)}^{20}, \tag{7a}$$

$$\begin{aligned}
C_{20}^{\text{Coupling}}(T) = &-\frac{T^{12}\Omega^6}{58060800}\left[\mathcal{M}_{(12,6)}^{20} + \Omega^2\mathcal{M}_{(12,8)}^{20}\right] \\
&+ \frac{T^{14}\Omega^6}{53648179200}\left[\mathcal{M}_{(14,6)}^{20} + \Omega^2\mathcal{M}_{(14,8)}^{20} + \Omega^4\mathcal{M}_{(14,10)}^{20}\right],
\end{aligned} \tag{7b}$$

$$C_{20}^{\text{Time}} = \frac{T^{12}\Omega^6}{116121600}\left[1 + \frac{T^2}{144}(\delta_f - \delta_0)^2 + \frac{T^4}{62208}(\delta_f - \delta_0)^4\right](\delta_f - \delta_0)^2\mathcal{M}_{(10,6)}^{20}. \tag{7c}$$

where

$$\begin{aligned}
\mathcal{M}_{(10,6)}^{20} &= \left[U_1^2(77U_1^2 - 340U_1\delta_{\text{avg}} + 375\delta_{\text{avg}}^2)\right]. \\
\mathcal{M}_{(12,6)}^{20} &= \left[2U_1^2(52U_1^4 - 368U_1^3\delta_{\text{avg}} + 1037U_1^2\delta_{\text{avg}}^2 - 1432U_1\delta_{\text{avg}}^3 + 831\delta_{\text{avg}}^4)\right], \\
\mathcal{M}_{(12,8)}^{20} &= \left\{U_1^2\left[593U_1^2 - 585U_1U_2 - 8(165U_1 - 308U_2)\delta_{\text{avg}} + 2544\delta_{\text{avg}}^2\right]\right\}, \\
\mathcal{M}_{(14,6)}^{20} &= \left[24U_1^2(121U_1^6 - 1170U_1^5\delta_{\text{avg}} + 4952U_1^4\delta_{\text{avg}}^2 - 11862U_1^3\delta_{\text{avg}}^3 \right. \\
&\quad \left. + 17112U_1^2\delta_{\text{avg}}^4 - 14322U_1\delta_{\text{avg}}^5 + 5565\delta_{\text{avg}}^6)\right], \\
\mathcal{M}_{(14,8)}^{20} &= \left\{U_1^2\left[30481U_1^4 - 44660U_1^3U_2 - 426030U_1^2U_2^2 - 28028U_1U_2^3 \right.\right. \\
&\quad - (208452U_1^3 - 60830U_2^3 - 249438U_1^2U_2 - 204960U_1U_2^2)\delta_{\text{avg}} \\
&\quad + (577781U_1^2 - 244923U_2^2 - 538804U_1U_2)\delta_{\text{avg}}^2 - (789348U_1 \\
&\quad \left.\left. - 457380U_2)\delta_{\text{avg}}^3 + 444780\delta_{\text{avg}}^4\right]\right\}, \\
\mathcal{M}_{(14,10)}^{20} &= \left\{U_1^2\left[80454U_1^2 + 13475U_2^2 - 197043U_1U_2 \right.\right. \\
&\quad \left.\left. - (317919U_1 - 137445U_2)\delta_{\text{avg}} + 103740\delta_{\text{avg}}^2\right]\right\}.
\end{aligned}$$

From Eq. (4) and Eq. (6), we observe that the components of correlation functions satisfy

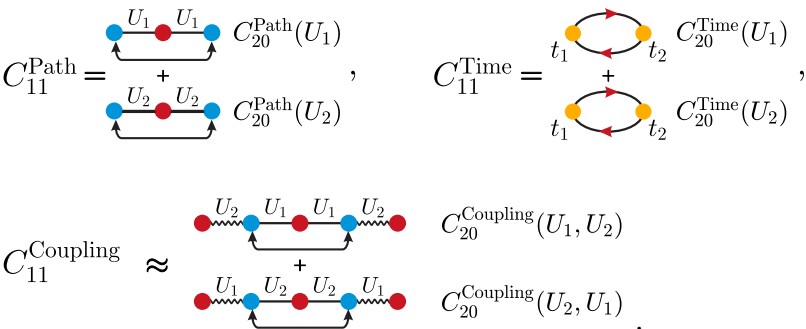

While minor discrepancies exist between $C_{11}^{\text{Coupling}}$ and $2 \times C_{20}^{\text{Coupling}}$ in specific coefficient terms, these deviations are negligible compared to the dominant contributions from $C_{11}^{\text{Path}}$ and $C_{11}^{\text{Time}}$. This is experimentally validated in Fig. 2, where under typical parameters the $C_{11}(T)$ curve completely overlaps with $2 \times C_{20}(T)$, even at extended time scales ($T = 1\mu s$). The observed relation $C_{11}(T) = 2 \times C_{20}(T)$ demonstrates that the correlation magnitude constitutes an algebraic summation of contributions from all shortest paths – a manifestation of path superposition principle. Notably, this principle remains unaffected by the presence of nonequivalent paths.

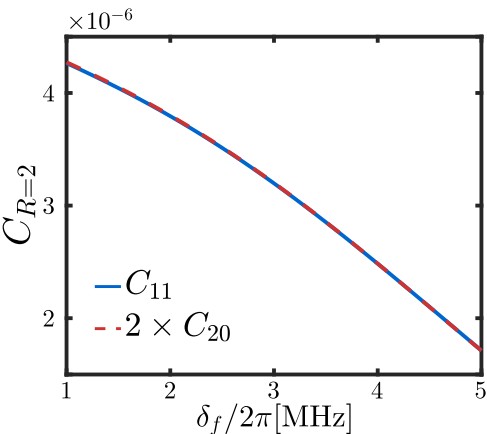

Figure 2: Superposition principle validation for the correlations in square lattice. Analytic verification of the next-nearest-neighbor correlation superposition with parameters referred to the experiment [10]: $T = 0.5\mu s$, $U_1/h = 2.8\,\text{MHz}$, $U_2/h = 1.4\,\text{MHz}$, $\Omega/(2\pi) = 0.8\,\text{MHz}$, $\delta_0/(2\pi) = -6\,\text{MHz}$, and $\delta_f/(2\pi) \in [1,5]\,\text{MHz}$. The near-perfect overlap between $C_{11}$ (blue solid line) and $2 \times C_{20}$ (red dashed line) confirms path contribution additivity.

### 3.2 Different lattice geometries

**$2 \times n$ lattice**    To investigate the influence of paths on the buildup of AF correlations, we begin by considering a $2 \times 12$ lattice array where $U_1 = 2U_2 = h \times 2.8\text{MHz}$ [see Fig. 3 (a)]. These nonuniform interactions give rise to two types of $C_{R=1}$, i.e., $C_{10,\text{square}}$ and $C_{01,\text{square}}$. Fig. 3 (c) shows that $|C_{10,\text{square}}| > |C_{01,\text{square}}|$ for the same $\delta_f$, indicating that in the quench process, a larger interaction $U_1$ tends to enhance the AF correlation. Meanwhile, our analytical results for $C_{R=1}$ are in strong quantitative agreement with the numerical results. Note that the absence of terms such as $C_{R=1}^{\text{Path}}, C_{R=1}^{\text{Coupling}}$, and $C_{R=1}^{\text{Time}}$ in the ME would prevent $C_{R=1}$ from matching the numerical results [see Appendix A]. Furthermore, we find that $C_{10,\text{square}}^{\text{Path/Time}}$ and $C_{01,\text{square}}^{\text{Path/Time}}$ share identical analytic forms, indicating that the shortest path and the mutual effects of the Hamiltonian at different times contribute equivalently to the AF correlations in the two structures. The critical difference between the two correlation functions stems from the nearest-neighbor coupling for the shortest path $C_{R=1}^{\text{Coupling}}$, which arises from two interdepentent factors: (i) the coupling strength $\Omega$ at the nearest-neighbor site to the shortest path, and (ii) the interaction $U$ between the shortest path and its adjacent sites. These differences directly reflect the geometric constraints of the respective lattices. Therefore, this highlights the importance of considering the coupling from the nearest-neighbor sites to the shortest path in the buildup of AF correlations across different lattice geometries and the necessity of including $C_{R=1}^{\text{Coupling}}$ in the ME.

In the calculation of $C_{R=2}$, ME provides the double realtion between $C_{11,\text{square}}$ and $C_{20,\text{chain1}}$

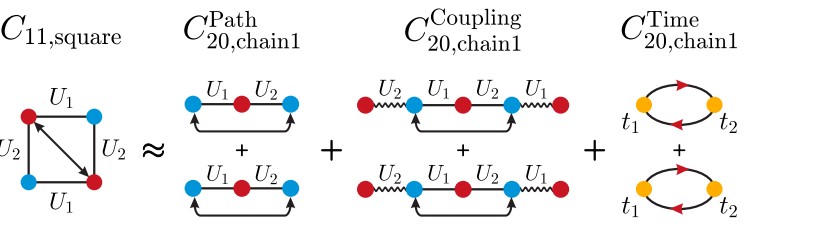

$$\tag{8}$$

where $C_{11,\text{square}}^{\text{Path/Time}} = 2 \times C_{20,\text{chain1}}^{\text{Path/Time}}$ and $C_{11,\text{square}}^{\text{Coupling}} \approx 2 \times C_{20,\text{chain1}}^{\text{Coupling}}$. This relation has been verified

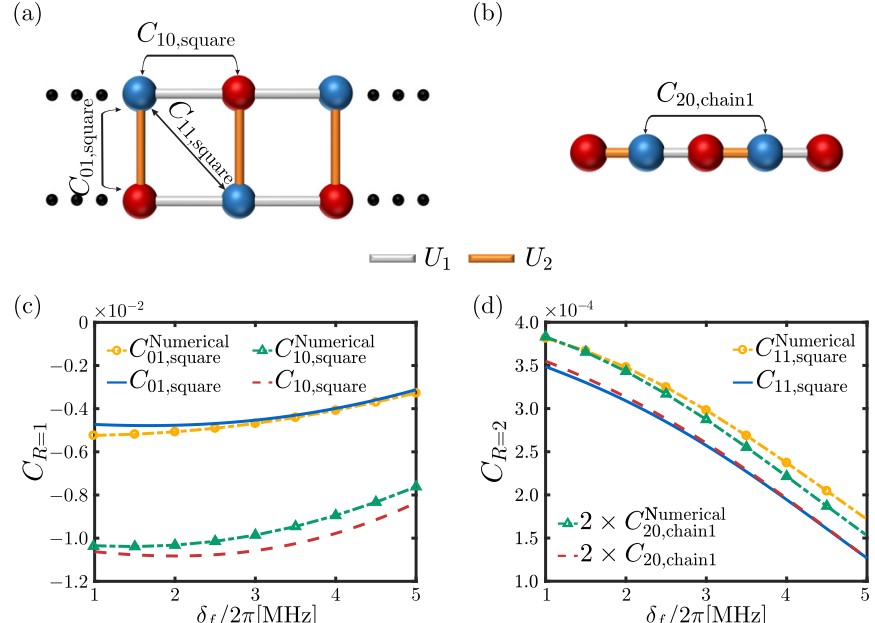

Figure 3: The buildup of antiferromagnetic correlation on $2 \times n$ lattice. Schematics of $2 \times n$ Rydberg array (a) and equivalent 1D chain capturing single-path contributions to $C_{11,\text{square}}$ (b). The nearest-neighbor correlation $C_{R=1}$ (c) and the next-nearest-neighbor correlations $C_{R=2}$ (d) as the function of $\delta_f$. The results of numerically solving Schrödinger equation for Hamiltonian (2) for $2 \times n$ lattice (yellow circles; green triangles) are compared with the analytic ones on the local lattice geometries (blue solid; red dashed).

through our numerical calculations [see Fig. 3 (d)]. Such agreement indicates that, despite the involvement of more complex paths for $C_{R=2}$ compared to $C_{R=1}$, the buildup of the AF correlation in a finitely large lattice array is still predominantly governed by the local structure around the correlated sites. The key factor behind the relation $C_{11,\text{square}} = 2 \times C_{20,\text{chain1}}$ originates from the fact that only one shortest path contributes to $C_{20,\text{chain1}}$, while there are two shortest paths for $C_{11,\text{square}}$. This implies the existence of a superposition law: the magnitude of the connected correlation function between two reference points is the algebraic sum of the contributions from all shortest paths.

**Cyclic lattice with a star**  To further investigate the effect of nonequivalent paths on spatial correlations, we consider the geometry of cyclic lattice with a star where $U_1/h = 2U_2/h = 2.8\,\text{MHz}$ and $U_3/h = 3.0\,\text{MHz}$ [see Fig. 4 (a)]. These nonuniform interactions give rise to three types of $C_{R=2}$: $C_{20,\text{cyclic}}$, $C_{20,\text{star:upper}}$, and $C_{20,\text{star:lower}}$. The results from ME show that $C_{20,\text{cyclic}}^{\text{Path}} = C_{20,\text{chain2a}}^{\text{Path}} + C_{20,\text{chain2b}}^{\text{Path}}$ and $C_{20,\text{cyclic}}^{\text{Time}} = C_{20,\text{chain2a}}^{\text{Time}} + C_{20,\text{chain2b}}^{\text{Time}}$, and $C_{20,\text{cyclic}}^{\text{Coupling}} \approx C_{20,\text{chian2a}}^{\text{Coupling}} + C_{20,\text{chain2b}}^{\text{Coupling}}$ [see Appendix A]. In a graphic description,

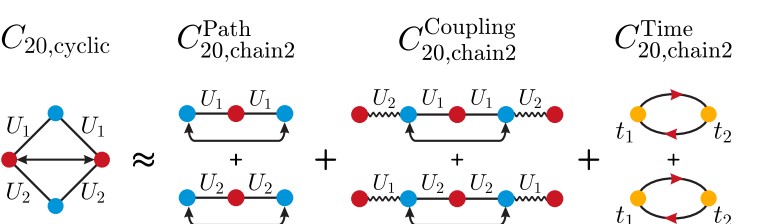

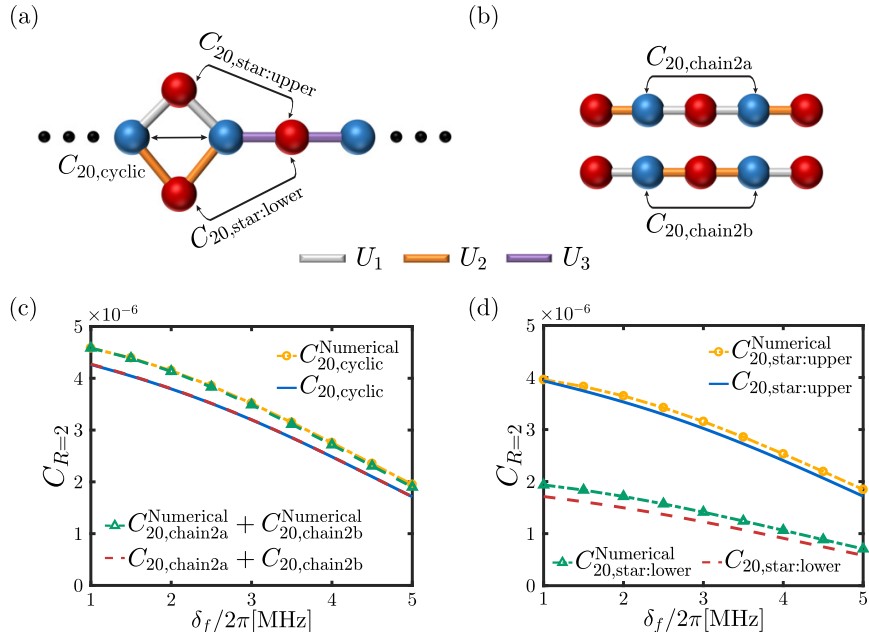

Figure 4: The buildup of antiferromagnetic correlation on cyclic lattice with a star. Schematics of cyclic lattice with a star (a) and equivalent 1D chains according to single-path contributions in the correlation $C_{20,\text{cyclic}}$ (b). The next-nearest-neighbor correlations $C_{20,\text{cyclic}}$ (c) and $C_{20,\text{star}}$ (d) as the function of $\delta_f$. The results of numerically solving Schrödinger equation for Hamiltonian (2) for cyclic lattice with a star (yellow circles; green triangles) are compared with the analytic ones on the local lattice geometries (blue solid; red dashed).

It demonstrates that the analytic results for $C_{20,\text{cyclic}}$ confirm the superposition law. By comparing $C_{20,\text{cyclic}}$ and its single-path contributions denoted by $C_{20,\text{chain2a}} + C_{20,\text{chain2b}}$, one can see that they are nearly overlapped [see Fig. 4 (c)]. Thus we verify both numerically and analytically the validity of superposition law, despite ME predictions for $C_{20,\text{cyclic}}$ show only small deviations ($< 10\%$) due to truncation error. In Fig. 4 (d), ME and numerical results for $C_{20,\text{star:upper}}$ agree within 4%, while for $C_{20,\text{star:lower}}$, deviations reach 14% under $U_1 = 2U_2$. This asymmetry stems from enhanced coupling: higher-order terms in ME become more important for the lower correlation, and neglecting these terms leads to a larger deviation from the exact results.

**Hexagonal and octagonal lattices** To extand superposition law to long-range correlations and reveal universal topology independence, we investigate the third-nearest-neighbor and the fourth-nearest-neighbor correlations in hexagonal and octagonal lattices respectively [see Fig. 5 (a) and (b)]. Due to the complicated and cumbersome expressions from ME, we illustrate the superposition law for these correlations using purely numerical simulations. We explore the single-path contributions through the numerical simulations in equivalent 1D-chain under the parameters used in hexagonal and octagonal lattices. After evoluting during the same time, we sum the correlations contributed from the single-path and compare respectively them with the correlations $C_{30,\text{hexagon}}$ and $C_{40,\text{octagon}}$.

Figure 5 (c) and (d) show $|C_{30,\text{hexagon}} - (C_{30,\text{chain a}} + C_{30,\text{chain b}})|/C_{30,\text{hexagon}} < 5\%$, and $|C_{40,\text{hexagon}} - (C_{40,\text{chain a}} + C_{40,\text{chain b}})|/C_{40,\text{hexagon}} < 4.5\%$ despite minor error growth at large $\delta_f$, demonstrating the validity of superposition law for longer correlations.

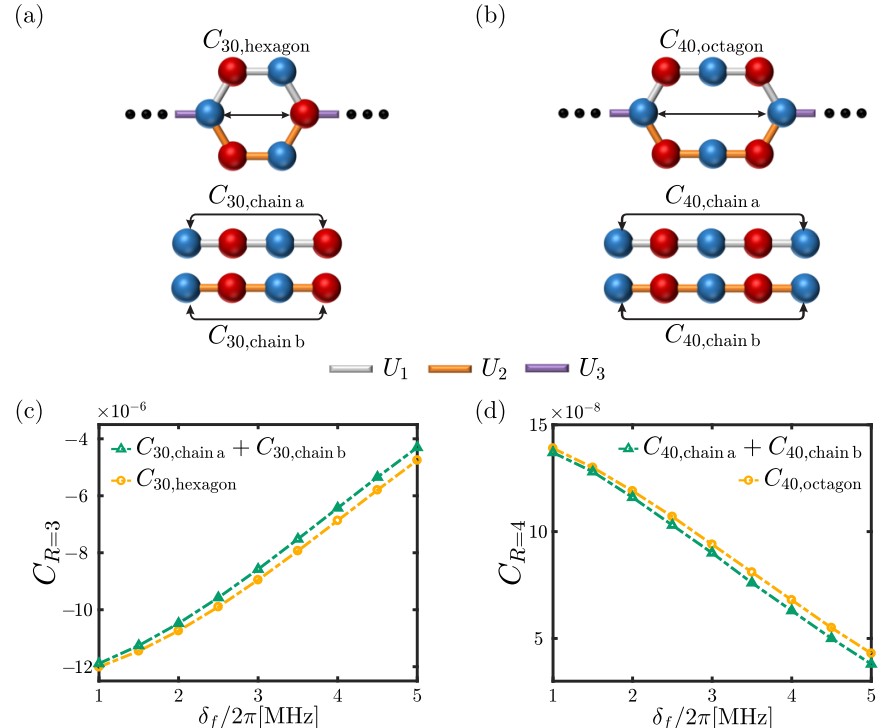

Figure 5: The buildup of antiferromagnetic correlation on hexagonal and octagonal lattices. Schematic descriptions of hexagon lattice (a), octagon lattice (b) and corresponding single-path contribution. The third-nearest-neighbor correlation $C_{30,\text{hexagon}}$ (c) and the fourth-nearest-neighbor correlation $C_{40,\text{octagon}}$ (d) as the function of $\delta_f$. The yellow dotted lines with circle and the green dotted lines with triangle are numerical results.

## 3.3 Different quench protocols

We demonstrate the universality of the superposition law under distinct quench styles using the cyclic lattice with a star as a benchmark.

**Quadratic quench** With detuning modulated as $\delta(t) = \frac{\delta_f - \delta_0}{T^2} t^2 + \delta_0$ ($t_{\text{sweep}} = 0.4\mu s$), ME yields ($\hbar = 1$):

$$\bar{H}_1 = \frac{\Omega}{2} \sum_i \sigma_i^x - \delta_{\text{avg}} \sum_i n_i + \sum_{\langle ij \rangle} U_{ij} n_i n_j \,,$$

and

$$\bar{H}_2 = \frac{\Omega}{24} (\delta_f - \delta_0) T \sum_i \sigma_i^y \,.$$

Remarkably, $\bar{H}_1$ and $\bar{H}_2$ maintain identical operator forms to the linear quench case, differing only in the averaged detuning (quadratic: $\frac{2\delta_0 + \delta_f}{3}$; linear: $\frac{\delta_0 + \delta_f}{2}$). This structural invariance leads to analytical similarity in correlation solutions. Fig. 6 (b) shows qualitative agreement between the analytic and numerical results for $C_{20,\text{cyclic}}$. Crucially, the respective overlaps of the numerical (yellow/green) and analytic (blue/red) results imply that the superposition law remains, i.e., $C_{20,\text{cyclic}} = C_{20,\text{chain2a}} + C_{20,\text{chain2b}}$.

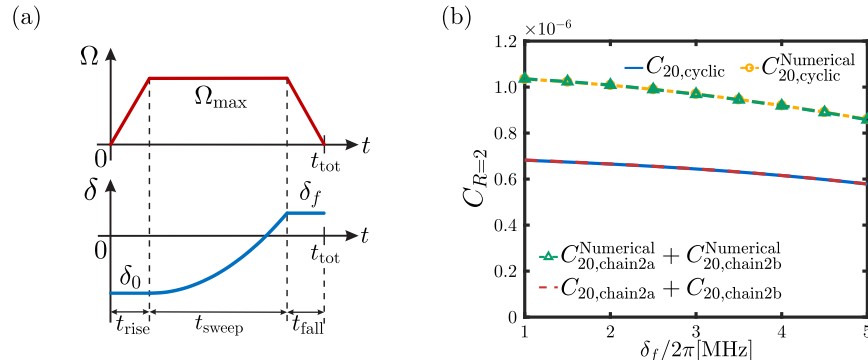

Figure 6: The buildup of antiferromagnetic correlation with quadratic quench process on cyclic lattice with a star. (a) The Rabi frequency $\Omega(t)$ and the detuning $\delta(t)$ are modulated in top and bottom curves, respectively. (b) The next-nearest-neighbor correlations $C_{20,\text{cyclic}}$ as the function of $\delta_f$.

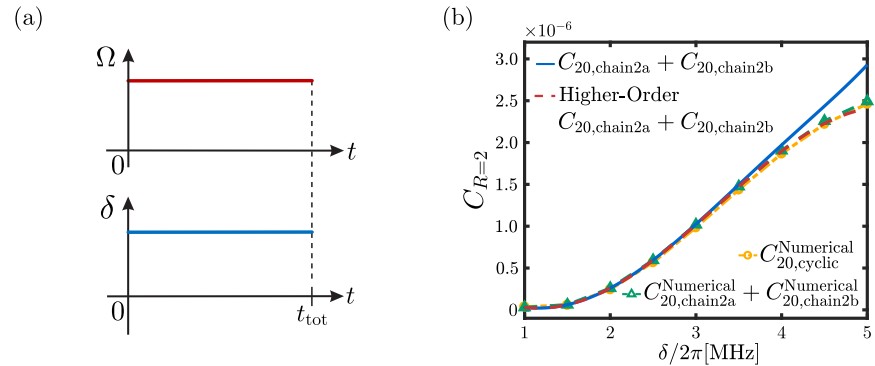

Figure 7: The buildup of antiferromagnetic correlation with sudden quench process on cyclic lattice with a star. (a) The Rabi frequency $\Omega$ and the detuning $\delta$ are modulated in top and bottom curves, respectively. (b) The next-nearest-neighbor correlations $C_{20,\text{cyclic}}$ as the function of $\delta$.

**Sudden quench** For instantaneous switching (constant $\Omega$, $\delta$ during evolution), we can derive $\bar{H}_1 = H$ and $\bar{H}_2 = 0$ using ME, which now simplifies to static expansion. Fig. 7 (b) reveals higher-order ME (red dashed) quantitatively matches the numerical results (yellow/green) although base solution $C_{20,\text{chain2}}$ (blue solid) deviates at large $\delta$. Superposition law prevails: path-sum predictions agree with full-lattice correlations.

The persistence of path superposition under linear (Fig. 4), quadratic (Fig. 6), and sudden (Fig. 7) quenches confirms its robustness against driving-protocol variations – a cornerstone for nonequilibrium quantum control.

# 4 Conclusion

In summary, our combined numerical and analytical study of AF correlation dynamics in driven Rydberg arrays reveals a fundamental organizing principle for nonequilibrium quantum systems. The AF correlation magnitude between lattice points at Manhattan distance $R$ (exemplified for $R = 2, 3, 4$) emerges as a coherent sum of amplitudes over all shortest paths, establishing a dynamical superposition principle that transcends path topology, lattice geometry, and quench protocol. Moreover, it also implies that probing large-scale complex systems – which

are computationally intractable – can be reduced to studying correlations in representative small-scale units. We further propose that the ME framework and the identified correlation buildup mechanism may extend to other strongly correlated systems, such as the Heisenberg model and Hubbard model, offering a potential pathway to decode their nonequilibrium dynamics. Moreover, investigating the persistence of the path superposition principle in periodically driven systems, for instance using Floquet Perturbation Theory [35], represents an important future research direction.

## Acknowledgments

We would like to thank Uwe R. Fischer and Jun-Hui Zheng for stimulating discussions.

**Funding information** This work is supported by the National Natural Science Foundation of China under Grants No.12075175 and No.11775178.

## A The analytic expressions for various geometries

According to the results of all kinds of local structures, we find a universal description for the standard terms: the denominator of all terms in these formulas can be simplified to $\Omega^p \delta^q U^m T^n$, where $p$, $q$, $m$, $n$ are integers and $n = p + q + m$. The interaction $U$ and detuning $\delta$ are variably involved in the most of terms, so we can extract the common factors that generally include $T$ and $\Omega$. Then we define a standard term as $\mathcal{M}^i_{(n,p)}$ for simply showing the analytic results, where $i$ refers to the label of various local structures and $n$ and $p$ are the power numbers of $T$ and $\Omega$, respectively. The graphic descriptions about these local structures are shown as follows. The solid lines between two reference balls label the shortest paths and the wave lines denote the coupling from the nearest-neighbor sites to the shortest paths.

There are two types of the nearest-neighbor correlation, $C_{10,\text{square}}$ and $C_{01,\text{square}}$. We label them as $a$ and $b$ respectively

$$C_{10,\text{square}} \qquad C_{10,\text{square}}^{\text{Path}} \qquad C_{10,\text{square}}^{\text{Coupling}} \qquad C_{10,\text{square}}^{\text{Time}}$$

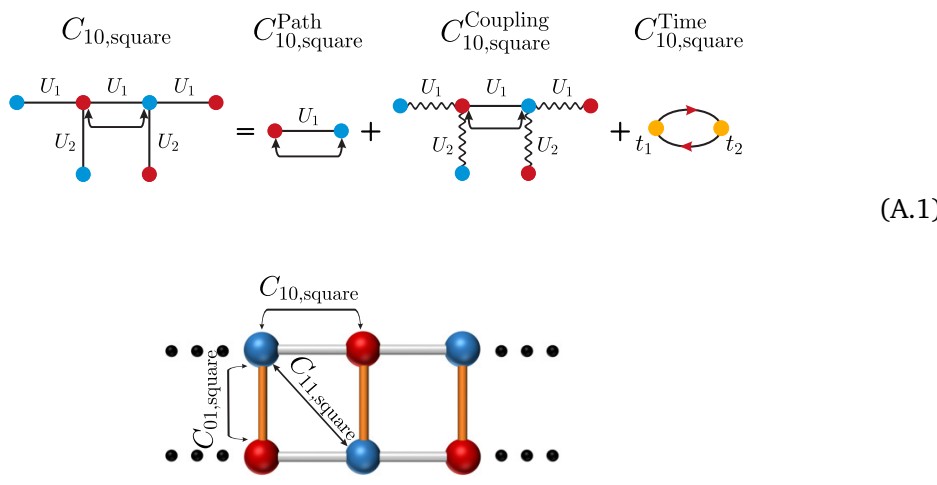

$$(\text{A.1})$$

Figure 8: $2 \times n$ *lattice* – Here, we give more details about the analytic expressions of $2 \times n$ lattice that derived from the ME.

the contribution from the shortest path is

$$C_{10,\text{square}}^{\text{Path}}(T) = -\frac{T^6\Omega^4}{288}\mathcal{M}_{(6,4)}^a,\tag{A.2}$$

where

$$\mathcal{M}_{(6,4)}^a = \left[U_1(U_1 - 3\delta_{\text{avg}})\right],$$

the one from the nearest-neighbor coupling for the shortest path is

$$C_{10,\text{square}}^{\text{Coupling}}(T) = \frac{T^8\Omega^4}{11520}\left[\mathcal{M}_{(8,4)}^a + \Omega^2\mathcal{M}_{(8,6)}^a\right] - \frac{T^{10}\Omega^4}{4838400}\left[\mathcal{M}_{(10,4)}^a - \Omega^2\mathcal{M}_{(10,6)}^a - \Omega^4\mathcal{M}_{(10,8)}^a\right],\tag{A.3}$$

where

$$
\begin{aligned}
\mathcal{M}_{(8,4)}^a &= \left[U_1(U_1^3 - 6U_1^2\delta_{\text{avg}} + 15U_1\delta_{\text{avg}}^2 - 18\delta_{\text{avg}}^3)\right],\\
\mathcal{M}_{(8,6)}^a &= \left[2U_1(2U_1 - 5U_2 - 18\delta_{\text{avg}})\right],\\
\mathcal{M}_{(10,4)}^a &= \left[2U_1(U_1 - 3\delta_{\text{avg}})(3U_1^4 - 18U_1^3\delta_{\text{avg}} + 55U_1^2\delta_{\text{avg}}^2 - 84U_1\delta_{\text{avg}}^3 + 85\delta_{\text{avg}}^4)\right],\\
\mathcal{M}_{(10,6)}^a &= \left[550U_1^4 + 3U_1^3(65U_2 - 388\delta_{\text{avg}}) + U_1^2(293U_2^2 - 945U_2\delta_{\text{avg}} - 246\delta_{\text{avg}}^2)\right.\\
&\qquad \left. + 5U_1(42U_2^3 - 209U_2^2\delta_{\text{avg}} + 376U_2\delta_{\text{avg}}^2 + 492\delta_{\text{avg}}^3)\right],\\
\mathcal{M}_{(10,8)}^a &= \left[U_1(674U_1 + 1505U_2 + 1950\delta_{\text{avg}})\right],
\end{aligned}
$$

and the one from the mutual effect of Hamiltonians at different times is

$$C_{10,\text{square}}^{\text{Time}}(T) = -\frac{T^8\Omega^4}{20736}\left[1 + \frac{T^2}{288}(\delta_f - \delta_0)^2\right](\delta_f - \delta_0)^2\mathcal{M}_{(6,4)}^a.\tag{A.4}$$

Considering the nearest-neighbor correlation in vertical direction,



$$\tag{A.5}$$

three parts are corresponded to expressions as follows

$$C_{01,\text{square}}^{\text{Path}}(T) = -\frac{T^6\Omega^4}{288}\mathcal{M}_{(6,4)}^b,\tag{A.6}$$

where

$$\mathcal{M}_{(6,4)}^b = \left[U_1(U_1 - 3\delta_{\text{avg}})\right].$$

$$C_{01,\text{square}}^{\text{Coupling}}(T) = \frac{T^8\Omega^4}{11520}\left[\mathcal{M}_{(8,4)}^b + \Omega^2\mathcal{M}_{(8,6)}^b\right] - \frac{T^{10}\Omega^4}{4838400}\left[\mathcal{M}_{(10,4)}^b - \Omega^2\mathcal{M}_{(10,6)}^b - \Omega^4\mathcal{M}_{(10,8)}^b\right],\tag{A.7}$$

where

$$\mathcal{M}^b_{(8,4)} = \left[ U_1(U_1^3 - 6U_1^2\delta_{\text{avg}} + 15U_1\delta_{\text{avg}}^2 - 18\delta_{\text{avg}}^3) \right],$$

$$\mathcal{M}^b_{(8,6)} = \left[ 2U_1(7U_1 - 10U_2 - 18\delta_{\text{avg}}) \right],$$

$$\mathcal{M}^b_{(10,4)} = \left[ 2U_1(U_1 - 3\delta_{\text{avg}})(3U_1^4 - 18U_1^3\delta_{\text{avg}} + 55U_1^2\delta_{\text{avg}}^2 - 84U_1\delta_{\text{avg}}^3 + 85\delta_{\text{avg}}^4) \right],$$

$$\mathcal{M}^b_{(10,6)} = \left[ -148U_1^4 + 2U_1^3(195U_2 + 413\delta_{\text{avg}}) + 2U_1^2(293U_2^2 - 945U_2\delta_{\text{avg}} - 1063\delta_{\text{avg}}^2) \right.$$
$$\left. + 10U_1(42U_2^3 - 209U_2^2\delta_{\text{avg}} + 376U_2\delta_{\text{avg}}^2 + 246\delta_{\text{avg}}^3) \right],$$

$$\mathcal{M}^b_{(10,8)} = \left[ U_1(-831U_1 + 3010U_2 + 1950\delta_{\text{avg}}) \right].$$

$$C^{\text{Time}}_{01,\text{square}}(T) = -\frac{T^8\Omega^4}{20736}\left[ 1 + \frac{T^2}{288}(\delta_f - \delta_0)^2 \right](\delta_f - \delta_0)^2 \mathcal{M}^b_{(6,4)}. \tag{A.8}$$

Compared these expressions, we find that the contribution from the shortest path $C^{\text{Path}}$ and the mutual effect of the Hamiltonian at different time $C^{\text{Time}}$ are identical in $C_{10,\text{square}}$ and $C_{01,\text{square}}$.

For the next-nearest-neighbor correlation $C_{11,\text{square}}$ that is labeled as $c$

$$\tag{A.9}$$

$$C^{\text{Path}}_{11,\text{square}}(T) = \frac{T^{10}\Omega^6}{4838400}\mathcal{M}^c_{(10,6)}, \tag{A.10}$$

where

$$\mathcal{M}^c_{(10,6)} = \left\{ U_1 U_2 \left[ 35(U_1^2 + U_2^2) + 238U_1 U_2 - 680(U_1 + U_2)\delta_{\text{avg}} + 1500\delta_{\text{avg}}^2 \right] \right\}.$$

$$C^{\text{Coupling}}_{11,\text{square}}(T) = -\frac{T^{12}\Omega^6}{58060800}\left[ \mathcal{M}^c_{(12,6)} - \Omega^2\mathcal{M}^c_{(12,8)} \right]$$
$$+ \frac{T^{14}\Omega^6}{53648179200}\left[ \mathcal{M}^c_{(14,6)} - \mathcal{M}^c_{(14,8)} - \mathcal{M}^c_{(14,10)} \right], \tag{A.11}$$

where

$$\mathcal{M}^c_{(12,6)} = \left\{ U_1 U_2 \left[ 12(U_1^4 + U_2^4) + 76(U_1^3 U_2 + U_1 U_2^3) + 32U_1^2 U_2^2 - 254(U_1^3 + U_2^3)\delta_{\text{avg}} \right.\right.$$
$$-482(U_1^2 U_2 + U_1 U_2^2)\delta_{\text{avg}} + 1183(U_1^2 + U_2^2)\delta_{\text{avg}}^2 + 1782U_1 U_2\delta_{\text{avg}}^2$$
$$\left.\left. - 2864(U_1 + U_2)\delta_{\text{avg}}^3 + 3324\delta_{\text{avg}}^4 \right] \right\},$$

$$\mathcal{M}^c_{(12,8)} = \left\{ U_1 U_2 \left[ 356(U_1^2 + U_2^2) - 280U_1 U_2 + 1144(U_1 + U_2)\delta_{\text{avg}} - 5088\delta_{\text{avg}}^2 \right] \right\},$$

$$\mathcal{M}^c_{(14,6)} = \left\{ U_1 U_2 \left[ 154(U_1^6 + U_2^6) + 1056(U_1^5 U_2 + U_1 U_2^5) - 3864(U_1^5 + U_2^5)\delta_{\text{avg}} \right.\right.$$
$$+26936(U_1^4 + U_2^4)\delta_{\text{avg}}^2 + 726(U_1^4 U_2^2 + U_1^2 U_2^4) - 10410(U_1^4 U_2 + U_1 U_2^4)\delta_{\text{avg}}$$
$$+1936U_1^3 U_2^3 - 13806(U_1^3 U_2^2 + U_1^2 U_2^3)\delta_{\text{avg}} + 58102(U_1^3 U_2 + U_1 U_2^3)\delta_{\text{avg}}^2$$
$$-103908(U_1^3 + U_2^3)\delta_{\text{avg}}^3 - 180780(U_1^2 U_2 + U_1 U_2^2)\delta_{\text{avg}}^3 + 241458(U_1^2 + U_2^2)\delta_{\text{avg}}^4$$
$$\left.\left. - 343728(U_1 + U_2)\delta_{\text{avg}}^5 + 67620U_1^2 U_2^2\delta_{\text{avg}}^2 + 338460U_1 U_2\delta_{\text{avg}}^4 + 267120\delta_{\text{avg}}^6 \right] \right\},$$

$$\mathcal{M}^c_{(14,8)} = \Big\{ U_1 U_2 \big[ 32879(U_1^4 + U_2^4) + 70950(U_1^3 U_2 + U_1 U_2^3) - 155848(U_1^3 + U_2^3)\delta_{\mathrm{avg}}$$
$$- 339944(U_1^2 U_2 + U_1 U_2^2)\delta_{\mathrm{avg}} + 192850(U_1^2 + U_2^2)\delta_{\mathrm{avg}}^2 + 331968(U_1 + U_2)\delta_{\mathrm{avg}}^3$$
$$+ 120142 U_1^2 U_2^2 + 298844 U_1 U_2 \delta_{\mathrm{avg}}^2 - 889560\delta_{\mathrm{avg}}^4 \big] \Big\},$$
$$\mathcal{M}^c_{(14,10)} = \Big\{ U_1 U_2 \big[ 143682(U_1^2 + U_2^2) - 94416(U_1 + U_2)\delta_{\mathrm{avg}} + 42372 U_1 U_2$$
$$- 622440\delta_{\mathrm{avg}}^2 \big] \Big\}.$$

$$C^{\mathrm{Time}}_{11,\mathrm{square}}(T) = \frac{T^{12}\Omega^6}{232243200}\left[ 1 + \frac{T^2}{144}(\delta_f - \delta_0)^2 + \frac{T^4}{62208}(\delta_f - \delta_0)^4 \right](\delta_f - \delta_0)^2 \mathcal{M}^c_{(10,6)}.$$
$$(A.12)$$

To verify analytically the validity of superposition principle in the $2 \times n$ lattice with nonuniform interaction, we choose $C_{20,\mathrm{chain1}}$ that refers to single-path contribution of $C_{11,\mathrm{square}}$, which is denoted as $d$

$$C_{20,\mathrm{chain1}} \qquad C^{\mathrm{Path}}_{20,\mathrm{chain1}} \qquad C^{\mathrm{Coupling}}_{20,\mathrm{chain1}} \qquad C^{\mathrm{Time}}_{20,\mathrm{chain1}}$$



$$(A.13)$$

where the contribution from the shortest path is

$$C^{\mathrm{Path}}_{20,\mathrm{chain1}}(T) = \frac{T^{10}\Omega^6}{9676800}\mathcal{M}^d_{(10,6)}, \qquad (A.14)$$

where

$$\mathcal{M}^d_{(10,6)} = \Big\{ U_1 U_2 \big[ 35(U_1^2 + U_2^2) + 238 U_1 U_2 - 680(U_1 + U_2)\delta_{\mathrm{avg}} + 1500\delta_{\mathrm{avg}}^2 \big] \Big\},$$

the one from the nearest-neighbor coupling for the shortest path is

$$C^{\mathrm{Coupling}}_{20,\mathrm{chain1}}(T) = -\frac{T^{12}\Omega^6}{116121600}\left[ \mathcal{M}^d_{(12,6)} - \Omega^2 \mathcal{M}^d_{(12,8)} \right]$$
$$+ \frac{T^{14}\Omega^6}{107296358400}\left[ \mathcal{M}^d_{(14,6)} - \Omega^2 \mathcal{M}^d_{(14,8)} - \Omega^4 \mathcal{M}^d_{(14,10)} \right], \qquad (A.15)$$

where

$$\mathcal{M}^d_{(12,6)} = \Big\{ U_1 U_2 \big[ 12(U_1^4 + U_2^4) + 76(U_1^3 U_2 + U_1 U_2^3) + 32 U_1^2 U_2^2 - 254(U_1^3 + U_2^3)\delta_{\mathrm{avg}}$$
$$- 482(U_1^2 U_2 + U_1 U_2^2)\delta_{\mathrm{avg}} + 1183(U_1^2 + U_2^2)\delta_{\mathrm{avg}}^2 + 1782 U_1 U_2 \delta_{\mathrm{avg}}^2$$
$$- 2864(U_1 + U_2)\delta_{\mathrm{avg}}^3 + 3324\delta_{\mathrm{avg}}^4 \big] \Big\},$$
$$\mathcal{M}^d_{(12,8)} = \Big\{ U_1 U_2 \big[ 356(U_1^2 + U_2^2) - 728 U_1 U_2 + 1144(U_1 + U_2)\delta_{\mathrm{avg}} - 5088\delta_{\mathrm{avg}}^2 \big] \Big\},$$
$$\mathcal{M}^d_{(14,6)} = \Big\{ U_1 U_2 \big[ 154(U_1^6 + U_2^6) + 1056(U_1^5 U_2 + U_1 U_2^5) - 3864(U_1^5 + U_2^5)\delta_{\mathrm{avg}}$$
$$+ 26936(U_1^4 + U_2^4)\delta_{\mathrm{avg}}^2 + 726(U_1^4 U_2^2 + U_1^2 U_2^4) - 10410(U_1^4 U_2 + U_1 U_2^4)\delta_{\mathrm{avg}}$$
$$+ 1936 U_1^3 U_2^3 - 13806(U_1^3 U_2^2 + U_1^2 U_2^3)\delta_{\mathrm{avg}} + 58102(U_1^3 U_2 + U_1 U_2^3)\delta_{\mathrm{avg}}^2$$
$$- 103908(U_1^3 + U_2^3)\delta_{\mathrm{avg}}^3 - 180780(U_1^2 U_2 + U_1 U_2^2)\delta_{\mathrm{avg}}^3 + 241458(U_1^2 + U_2^2)\delta_{\mathrm{avg}}^4$$
$$- 343728(U_1 + U_2)\delta_{\mathrm{avg}}^5 + 67620 U_1^2 U_2^2 \delta_{\mathrm{avg}}^2 + 338460 U_1 U_2 \delta_{\mathrm{avg}}^4 + 267120\delta_{\mathrm{avg}}^6 \big] \Big\},$$



$$\mathcal{M}^d_{(14,8)} = \Big\{U_1 U_2 \Big[32879(U_1^4 + U_2^4) + 25949(U_1^3 U_2 + U_1 U_2^3) - 155848(U_1^3 + U_2^3)\delta_{avg}$$
$$- 150928(U_1^2 U_2 + U_1 U_2^2)\delta_{avg} + 192850(U_1^2 + U_2^2)\delta_{avg}^2 + 331968(U_1 + U_2)\delta_{avg}^3$$
$$+ 51964 U_1^2 U_2^2 + 26192 U_1 U_2 \delta_{avg}^2 - 889560\delta_{avg}^4\Big]\Big\},$$
$$\mathcal{M}^d_{(14,10)} = \Big\{U_1 U_2 \Big[143682(U_1^2 + U_2^2) - 94416(U_1 + U_2)\delta_{avg} - 135036 U_1 U_2$$
$$- 622440\delta_{avg}^2\Big]\Big\},$$

and the one from the mutual effect of Hamiltonians at different times,

$$C^{\text{Time}}_{20,\text{chain1}}(T) = \frac{T^{12}\Omega^6}{464486400}\left[1 + \frac{T^2}{144}(\delta_f - \delta_0)^2 + \frac{T^4}{62208}(\delta_f - \delta_0)^4\right](\delta_f - \delta_0)^2 \mathcal{M}^d_{(10,6)}.$$
(A.16)

For $C_{20,\text{chain1}}$ and $C_{11,\text{square}}$, we find that $C^{\text{Path}}_{11,\text{square}}$ and $C^{\text{Time}}_{11,\text{square}}$ are certainly doubled $C^{\text{Path}}_{20,\text{chain1}}$ and $C^{\text{Time}}_{20,\text{chain1}}$ accordingly. For the second term, $C^{\text{Coupling}}_{11,\text{square}}$ is approximately twice $C^{\text{Coupling}}_{20,\text{chain1}}$ except there are slightly difference of coefficients in few terms.

Figure 9 shows that our analytic results, $C_{10,\text{square}}$, $C_{01,\text{square}}$ and $C_{11,\text{square}}$ in accordance with the local lattice arrays match well the exact numerical results for the $2 \times n$ lattice. It implies that in a short-time regime, the buildup of the local correlation in the large-scale lattice system is determined predominantly by the local lattice geometry around the correlated sites. Meanwhile, with increasing $\delta_f$, the antiferromagnetic (AF) correlations are suppressed gradually. Since $C^{\text{Path}}_R$ only agrees with the tendency of the numerical results, the leading approximation of ME can not describe precisely the AF correlation. When $C^{\text{Coupling}}_R$ is taken into account, the variation of ME result is identical to the exact numerical results, demonstrating $C^{\text{Path}}_R + C^{\text{Coupling}}_R$ can describe qualitatively the buildup of the AF correlation in the nonequilibrium dynamics. Yet when we consider the term of $C^{\text{Time}}_R$, the ME results can quantitatively describe the connected correlation function. Also the analytic expression shows for fixed $T$, $\Omega$, $\delta_f$, the larger $U$, the larger $C_{R=1}$, so the AF correlation betweem horizontal sites is larger than the one betweem vertical sites.

We first consider three terms in $C_{20,\text{cyclic}}$, which is labeled as $e$

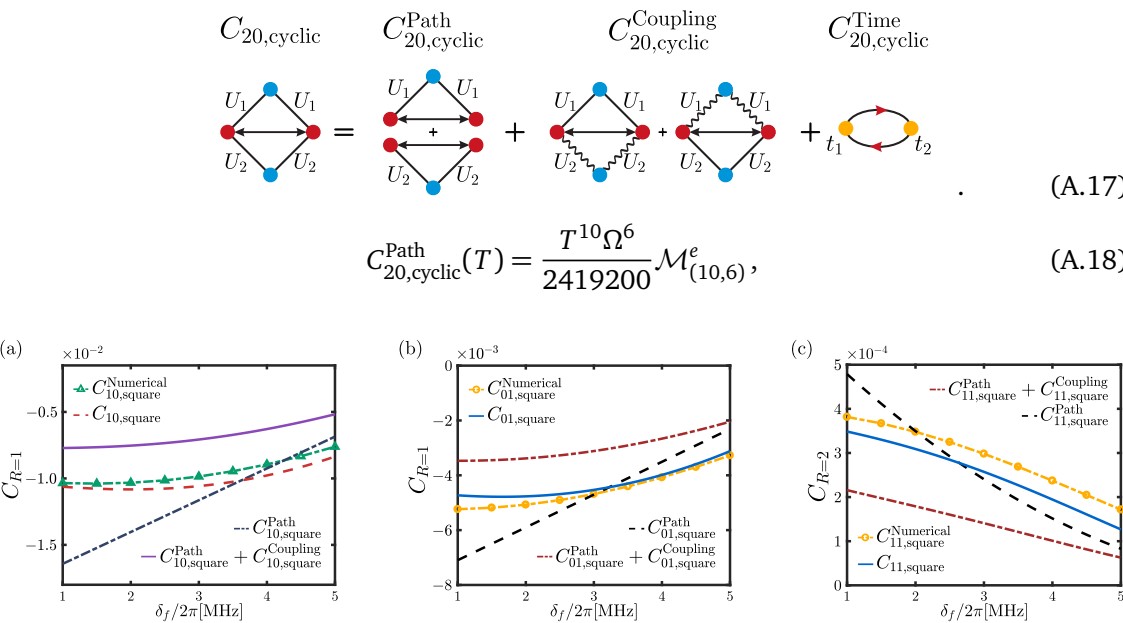

(A.17)

$$C^{\text{Path}}_{20,\text{cyclic}}(T) = \frac{T^{10}\Omega^6}{2419200}\mathcal{M}^e_{(10,6)},$$
(A.18)

Figure 9: The nearest-neighbor correlations $C_{10,\text{square}}$ (a) and $C_{01,\text{square}}$ (b), and the next-nearest-neighbor correlation $C_{11,\text{square}}$ (c) as the function of $\delta_f$ in $2 \times n$ lattice.

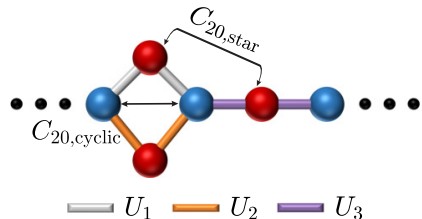

Figure 10: *Cyclic lattice with a star* – There are specific analytic expression of cyclic lattice with a star. We mainly consider two types of the next-nearest-neighbor correlation, i.e., $C_{20,\text{cyclic}}$ and $C_{20,\text{star}}$ in this model.

where

$$\mathcal{M}^e_{(10,6)} = \Big[ 77(U_1^4 + U_2^4) - 340(U_1^3 + U_2^3)\delta_{\text{avg}} + 375(U_1^2 + U_2^2)\delta^2_{\text{avg}} \Big].$$

$$
\begin{aligned}
C_{20,\text{cyclic}}^{\text{Coupling}}(T) = &-\frac{T^{12}\Omega^6}{58060800}\Big[ \mathcal{M}^e_{(12,6)} + \Omega^2 \mathcal{M}^e_{(12,8)} \Big] \\
&+ \frac{T^{14}\Omega^6}{53648179200}\Big[ \mathcal{M}^e_{(14,6)} + \Omega^2 \mathcal{M}^e_{(14,8)} + \Omega^4 \mathcal{M}^e_{(14,10)} \Big],
\end{aligned}
\tag{A.19}
$$

where

$$
\begin{aligned}
\mathcal{M}^e_{(12,6)} = &\Big[ 104(U_1^6 + U_2^6) - 736(U_1^5 + U_2^5)\delta_{\text{avg}} + 2074(U_1^4 + U_2^4)\delta^2_{\text{avg}} - 2864(U_1^3 + U_2^3)\delta^3_{\text{avg}} \\
&+ 1662(U_1^2 + U_2^2)\delta^4_{\text{avg}} \Big], \\
\mathcal{M}^e_{(12,8)} = &\Big\{ 593(U_1^4 + U_2^4) - 585(U_1^3 U_2 + U_1 U_2^3) - 448 U_1^2 U_2^2 \\
&- 88(U_1 + U_2)\Big[ 28(U_1^2 + U_2^2) - 43 U_1 U_2 \Big]\delta_{\text{avg}} + 2544(U_1^2 + U_2^2)\delta^2_{\text{avg}} \Big\}, \\
\mathcal{M}^e_{(14,6)} = &\Big\{ 24\Big[ 121(U_1^8 + U_2^8) - 1170(U_1^7 + U_2^7)\delta_{\text{avg}} + 4952(U_1^6 + U_2^6)\delta^2_{\text{avg}} - 11862(U_1^5 \\
&+ U_2^5)\delta^3_{\text{avg}} + 17112(U_1^4 + U_2^4)\delta^4_{\text{avg}} - 14322(U_1^3 + U_2^3)\delta^5_{\text{avg}} + 5565(U_1^2 + U_2^2)\delta^6_{\text{avg}} \Big] \Big\}, \\
\mathcal{M}^e_{(14,8)} = &\Big\{ 11\Big[ 2771(U_1^6 + U_2^6) - 4060(U_1^5 U_2 + U_1 U_2^5) - 7893(U_1^4 U_2^2 + U_1^2 U_2^4) \\
&- 11436 U_1^3 U_2^3 \Big] - 6(U_1 + U_2)\Big[ 34742(U_1^4 + U_2^4) - 76315(U_1^3 U_2 + U_1 U_2^3) \\
&+ 514 U_1^2 U_2^2 \Big]\delta_{\text{avg}} + \Big[ 577781(U_1^4 + U_2^4) - 538804(U_1^3 U_2 + U_1 U_2^3) \\
&- 762498 U_1^2 U_2^2 \Big]\delta^2_{\text{avg}} - 84(U_1 + U_2)\Big[ 9397(U_1^2 + U_2^2) - 14842 U_1 U_2 \Big]\delta^3_{\text{avg}} \\
&+ 444780(U_1^2 + U_2^2)\delta^4_{\text{avg}} \Big\}, \\
\mathcal{M}^e_{(14,10)} = &\Big\{ 33\Big[ 2438(U_1^4 + U_2^4) - 5971(U_1^3 U_2 + U_1 U_2^3) - 2926 U_1^2 U_2^2 \Big] \\
&- 21(U_1 + U_2)\Big[ 15139(U_1^2 + U_2^2) - 34774 U_1 U_2 \Big]\delta_{\text{avg}} + 311220(U_1^2 + U_2^2)\delta^2_{\text{avg}} \Big\}.
\end{aligned}
$$

$$C_{20,\text{cyclic}}^{\text{Time}}(T) = \frac{T^{12}\Omega^6}{116121600}\left[ 1 + \frac{T^2}{144}(\delta_f - \delta_0)^2 + \frac{T^4}{62208}(\delta_f - \delta_0)^4 \right](\delta_f - \delta_0)^2 \mathcal{M}^e_{(10,6)}.$$

$$\tag{A.20}$$

To analyze the superposition principle in cyclic lattice with a star, we calculate the one of the shortest paths in $C_{20,\text{cyclic}}$, which is named $C_{20,\text{chain2}}$ and denotes as $f$

$$\tag{A.21}$$

$$C_{20,\text{chain2}}^{\text{Path}}(T) = \frac{T^{10}\Omega^6}{2419200}\mathcal{M}_{(10,6)}^f, \tag{A.22}$$

where

$$\mathcal{M}_{(10,6)}^f = \left[ U_1^2(77U_1^2 - 340U_1\delta_{\text{avg}} + 375\delta_{\text{avg}}^2) \right].$$

$$
\begin{aligned}
C_{20,\text{chain2}}^{\text{Coupling}}(T) = &-\frac{T^{12}\Omega^6}{58060800}\left[ \mathcal{M}_{(12,6)}^f + \Omega^2\mathcal{M}_{(12,8)}^f \right] \\
&+ \frac{T^{14}\Omega^6}{53648179200}\left[ \mathcal{M}_{(14,6)}^f + \Omega^2\mathcal{M}_{(14,8)}^f + \Omega^4\mathcal{M}_{(14,10)}^f \right]
\end{aligned} \tag{A.23}
$$

where

$$
\begin{aligned}
\mathcal{M}_{(12,6)}^f &= \left[ 2U_1^2(52U_1^4 - 368U_1^3\delta_{\text{avg}} + 1037U_1^2\delta_{\text{avg}}^2 - 1432U_1\delta_{\text{avg}}^3 + 831\delta_{\text{avg}}^4) \right], \\
\mathcal{M}_{(12,8)}^f &= \left\{ U_1^2\left[ 593U_1^2 - 585U_1U_2 - 8(165U_1 - 308U_2)\delta_{\text{avg}} + 2544\delta_{\text{avg}}^2 \right] \right\}, \\
\mathcal{M}_{(14,6)}^f &= \left[ 24U_1^2(121U_1^6 - 1170U_1^5\delta_{\text{avg}} + 4952U_1^4\delta_{\text{avg}}^2 - 11862U_1^3\delta_{\text{avg}}^3 + 17112U_1^2\delta_{\text{avg}}^4 \right. \\
&\qquad\left. - 14322U_1\delta_{\text{avg}}^5 + 5565\delta_{\text{avg}}^6) \right], \\
\mathcal{M}_{(14,8)}^f &= \left[ U_1^2(30481U_1^4 - 44660U_1^3U_2 - 28028U_1U_2^3 - 208452U_1^3\delta_{\text{avg}} + 60830U_2^3\delta_{\text{avg}} \right. \\
&\qquad - 426030U_1^2U_2^2 + 249438U_1^2U_2\delta_{\text{avg}} + 204960U_1U_2^2\delta_{\text{avg}} + 577781U_1^2\delta_{\text{avg}}^2 \\
&\qquad - 244923U_2^2\delta_{\text{avg}}^2 - 538804U_1U_2\delta_{\text{avg}}^2 - 789348U_1\delta_{\text{avg}}^3 + 457380U_2\delta_{\text{avg}}^3 \\
&\qquad\left. + 444780\delta_{\text{avg}}^4) \right], \\
\mathcal{M}_{(14,10)}^f &= \left[ U_1^2(80454U_1^2 + 13475U_2^2 - 197043U_1U_2 - (317919U_1 - 137445U_2)\delta_{\text{avg}} \right. \\
&\qquad\left. + 103740\delta_{\text{avg}}^2) \right].
\end{aligned}
$$

$$C_{20,\text{chain2}}^{\text{Time}} = \frac{T^{12}\Omega^6}{116121600}\left[ 1 + \frac{T^2}{144}(\delta_f - \delta_0)^2 + \frac{T^4}{62208}(\delta_f - \delta_0)^4 \right](\delta_f - \delta_0)^2\mathcal{M}_{(10,6)}^f. \tag{A.24}$$

We define this correlation that central interaction is $U_1$ as $C_{20,\text{chain2a}}$. It is the one of shortest path contribution of $C_{20,\text{cyclic}}$. The contributions from another shortest path share identical correlation function forms while its central interaction is $U_2$, which is definned as $C_{20,\text{chain2b}}$. A comparison between $C_{20,\text{cyclic}}$ and the dual-path superposition ($C_{20,\text{chain2a}} + C_{20,\text{chain2b}}$) reveals that $C^{\text{Path}}$ and $C^{\text{Time}}$ are strictly identical, while $C^{\text{Coupling}}$ shows near equivalence with only minor discrepancies in the coefficients of few terms.

In the main content, we also discuss the robustness of the superposition law in cyclic lattice with a star under sudden quench process. In sudden quench, the second-order of ME is zero, and the first-order term only is considered. We find the deviation appears when $\delta$ becomes larger, so the higher-order expansion $\mathcal{M}_{(16)}^f$ in $C_{20,\text{chain2}}^{\text{Coupling}}$ is added, i.e.,

$$
\begin{aligned}
C_{20,\text{chain2}}^{\text{Coupling}}(T) = &-\frac{T^{12}\Omega^6}{58060800}\left[ \mathcal{M}_{(12,6)}^f + \Omega^2\mathcal{M}_{(12,8)}^f \right] \\
&+ \frac{T^{14}\Omega^6}{53648179200}\left[ \mathcal{M}_{(14,6)}^f + \Omega^2\mathcal{M}_{(14,8)}^f + \Omega^4\mathcal{M}_{(14,10)}^f \right] \\
&- \frac{T^{16}\Omega^6}{669529276416000}\left[ \mathcal{M}_{(16,6)}^f + \Omega^2\mathcal{M}_{(16,8)}^f + \Omega^4\mathcal{M}_{(16,10)}^f + \Omega^4\mathcal{M}_{(16,12)}^f \right]
\end{aligned} \tag{A.25}
$$

$$\mathcal{M}^f_{(16,6)} = \Big[48U_1^2(15535U_1^8 - 189784U_1^7\delta_{\text{avg}} + 1055262U_1^6\delta_{\text{avg}}^2 - 3502012U_1^5\delta_{\text{avg}}^3$$
$$+ 7618415U_1^4\delta_{\text{avg}}^4 - 11190036U_1^3\delta_{\text{avg}}^5 + 10925798U_1^2\delta_{\text{avg}}^6 - 6566960U_1\delta_{\text{avg}}^7$$
$$+ 1900758\delta_{\text{avg}}^8)\Big],$$

$$\mathcal{M}^f_{(16,8)} = U_1^2\Big[11341720U_1^6 - 22104264U_1^5U_2 - 34120294U_1^4U_2^2 - 33849764U_1^3U_2^3$$
$$- 18385302U_1^2U_2^4 - 7679360U_1U_2^5 - (107170848U_1^5 - 185405664U_1^4U_2$$
$$- 261040704U_1^3U_2^2 - 207519072U_1^2U_2^3 - 93857952U_1U_2^4 - 16328000U_2^5)\delta_{\text{avg}}$$
$$+ (451005016U_1^4 - 686392744U_1^3U_2 - 813247584U_1^2U_2^2 - 490813872U_1U_2^3$$
$$- 117018336U_2^4)\delta_{\text{avg}}^2 - (1083894192U_1^3 - 1405791120U_1^2U_2 - 1276827488U_1U_2^2$$
$$- 433703552U_2^3)\delta_{\text{avg}}^3 + (1573046512U_1^2 - 1617105680U_1U_2 - 852850320U_2^2)\delta_{\text{avg}}^4$$
$$- (1321518528U_1 - 860741568U_2)\delta_{\text{avg}}^5 + 506592576\delta_{\text{avg}}^6\Big],$$

$$\mathcal{M}^f_{(16,10)} = U_1^2\Big[48782565U_1^4 - 197318290U_1^3U_2 - 137542587U_1^2U_2^2 - 49352472U_1U_2^3$$
$$+ 50450400U_2^4 - (329050000U_1^3 - 1081513632U_1^2U_2 - 680292464U_1U_2^2$$
$$- 42525080U_2^3)\delta_{\text{avg}} + (929508952U_1^2 - 2327080592U_1U_2 - 707810760U_2^2)\delta_{\text{avg}}^2$$
$$- (1306218672U_1 - 1919036016U_2)\delta_{\text{avg}}^3 + 739476000\delta_{\text{avg}}^4\Big],$$

$$\mathcal{M}^f_{(16,12)} = U_1^2\Big[90884976U_1^2 - 430244256U_1U_2 + 193393200U_2^2 - (345319584U_1$$
$$- 843841440U_2)\delta_{\text{avg}} + 324119808\delta_{\text{avg}}^2\Big].$$

$C_{20,\text{star}}$ is marked as $g$,



$$\tag{A.26}$$

$$C^{\text{Path}}_{20,\text{star}}(T) = \frac{T^{10}\Omega^6}{9676800}\mathcal{M}^g_{(10,6)}, \tag{A.27}$$

where

$$\mathcal{M}^g_{(10,6)} = \Big\{U_1U_2\Big[35(U_1^2 + U_2^2) + 238U_1U_2 - 680(U_1 + U_2)\delta_{\text{avg}} + 1500\delta_{\text{avg}}^2\Big]\Big\}.$$

$$C^{\text{Coupling}}_{20,\text{star}}(T) = -\frac{T^{12}\Omega^6}{232243200}\Big[\mathcal{M}^g_{(12,6)} + \Omega^2\mathcal{M}^g_{(12,8)}\Big]$$
$$+ \frac{T^{14}\Omega^6}{214592716800}\Big[\mathcal{M}^g_{(14,6)} + \Omega^2\mathcal{M}^g_{(14,8)} + \Omega^4\mathcal{M}^g_{(14,10)}\Big], \tag{A.28}$$

where

$$
\begin{aligned}
\mathcal{M}^g_{(12,6)} = {}& U_1 U_2 \Big\{ 8 \big[ 3(U_1^4 + U_2^4) + 19(U_1^3 U_2 + U_1 U_2^3) + 8 U_1^2 U_2^2 \big] \\
& - 4(U_1 + U_2)\big[ 127(U_1^2 + U_2^2) + 114 U_1 U_2 \big] \delta_{\text{avg}} + 2\big[ 1183(U_1^2 + U_2^2) \\
& + 1782 U_1 U_2 \big] \delta_{\text{avg}}^2 - 5728(U_1 + U_2)\delta_{\text{avg}}^3 + 6648 \delta_{\text{avg}}^4 \Big\}, \\
\mathcal{M}^g_{(12,8)} = {}& U_1 U_2 \Big\{ 293(U_1^2 + U_2^2) - 126(U_1 U_3 + U_2 U_3) + 1786 U_1 U_2 \\
& - 112\big[ 44(U_1 + U_2) - 5U_3 \big] \delta_{\text{avg}} + 10176 \delta_{\text{avg}}^2 \Big\}, \\
\mathcal{M}^g_{(14,6)} = {}& 4 U_1 U_2 \Big\{ 77(U_1^6 + U_2^6) + 528(U_1^5 U_2 + U_1 U_2^5) + 363(U_1^4 U_2^2 + U_1^2 U_2^4) + 968 U_1^3 U_2^3 \\
& - \big[ 1932(U_1^5 + U_2^5) + 5205(U_1^4 U_2 + U_1 U_2^4) + 6903(U_1^3 U_2^2 + U_1^2 U_2^3) \big] \delta_{\text{avg}} \\
& + \big[ 13468(U_1^4 + U_2^4) + 29051(U_1^3 U_2 + U_1 U_2^3) + 33810 U_1^2 U_2^2 \big] \delta_{\text{avg}}^2 \\
& - \big[ 51954(U_1^3 + U_2^3) + 90390(U_1^2 U_2 + U_1 U_2^2) \big] \delta_{\text{avg}}^3 + \big[ 120729(U_1^2 + U_2^2) \\
& + 169230 U_1 U_2 \big] \delta_{\text{avg}}^4 - 171864(U_1 + U_2)\delta_{\text{avg}}^5 + 133560 \delta_{\text{avg}}^6 \Big\}, \\
\mathcal{M}^g_{(14,8)} = {}& U_1 U_2 \Big\{ 7700(U_1^4 + U_2^4) + 43549(U_1^3 U_2 + U_1 U_2^3) - 4235 U_1^3 U_3 - 4158 U_1 U_3^3 \\
& + 19426 U_1^2 U_2^2 - 4851 U_1^2 U_3^2 - 9625(U_1^2 U_2 + U_1 U_2^2)U_3 - 13134 U_1 U_2 U_3^2 \\
& - \big[ 142604 U_1^3 + 274300(U_1^2 U_2 + U_1 U_2^2) - 43547 U_1^2 U_3 - 48930 U_1 U_3^2 \\
& - 69818 U_1 U_2 U_3 \big] \delta_{\text{avg}} + (647717 U_1^2 + 1015690 U_1 U_2 - 161700 U_1 U_3)\delta_{\text{avg}}^2 \\
& - 1578696 U_1 \delta_{\text{avg}}^3 - 7\big[ 11 U_2(55 U_2^2 U_3 + 63 U_2 U_3^2 + 54 U_3^3) - (20372 U_2^3 \\
& - 6221 U_2^2 U_3 - 6990 U_2 U_3^2 - 2520 U_3^3)\delta_{\text{avg}} + (92531 U_2^2 - 23100 U_2 U_3 \\
& - 14844 U_3^2)\delta_{\text{avg}}^2 - 24(9397 U_2 - 1475 U_3)\delta_{\text{avg}}^3 + 254160 \delta_{\text{avg}}^4 \big] \Big\}, \\
\mathcal{M}^g_{(14,10)} = {}& U_1 U_2 \Big\{ 7\big[ 5973(U_1^2 + U_2^2) - 6248 U_3(U_1 + U_2) \big] + 238194 U_1 U_2 \\
& - 7\big[ 90834(U_1 + U_2) - 27300 U_3 \big] \delta_{\text{avg}} + 1244880 \delta_{\text{avg}}^2 \Big\}.
\end{aligned}
$$

$$
C^{\text{Time}}_{20,\text{star}}(T) = \frac{T^{12}\Omega^6}{464486400}\left[ 1 + \frac{T^2}{144}(\delta_f - \delta_0)^2 + \frac{T^4}{62208}(\delta_f - \delta_0)^4 \right](\delta_f - \delta_0)^2 \mathcal{M}^g_{(10,6)}. \tag{A.29}
$$

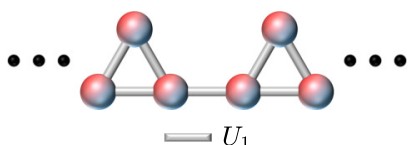

Figure 11: *Triangular lattice* – The analytic results of triangular lattice that derived from the ME show as follows. The expressions of the nearest-neighbor correlation betweem any two atoms in the triangle are identical. We denote this model as *h*.

As mentioned above, $C_{20,\mathrm{star}}^{\mathrm{Path}}$ and $C_{20,\mathrm{star}}^{\mathrm{Time}}$ are identical respectively with $C_{20,\mathrm{chain1}}^{\mathrm{Path}}$ and $C_{20,\mathrm{chain1}}^{\mathrm{Time}}$ due to the same shortest paths of them.

$$C_{R=1,\mathrm{triangle}} \qquad C_{R=1,\mathrm{triangle}}^{\mathrm{Path}} \qquad C_{R=1,\mathrm{triangle}}^{\mathrm{Coupling}} \qquad C_{R=1,\mathrm{triangle}}^{\mathrm{Time}}$$

$$(A.30)$$

where the contribution from the shortest path is

$$C_{R=1,\mathrm{triangle}}^{\mathrm{Path}}(T) = -\frac{T^6 \Omega^4}{288} \mathcal{M}_{(6,4)}^h, \tag{A.31}$$

where

$$\mathcal{M}_{(6,4)}^h = \left[ U_1(U_1 - 3\delta_{\mathrm{avg}}) \right],$$

the one from the nearest-neighbor coupling is

$$C_{R=1,\mathrm{triangle}}^{\mathrm{Coupling}}(T) = \frac{T^8 \Omega^4}{11520} \left[ \mathcal{M}_{(8,4)}^h + \Omega^2 \mathcal{M}_{(8,4)}^h \right] - \frac{T^{10} \Omega^4}{2419200} \left[ \mathcal{M}_{(10,4)}^h - \Omega^2 \mathcal{M}_{(10,6)}^h - \Omega^4 \mathcal{M}_{(10,8)}^h \right], \tag{A.32}$$

where

$$
\begin{aligned}
\mathcal{M}_{(8,4)}^h &= \left[ U_1(U_1^3 - 6U_1^2 \delta_{\mathrm{avg}} + 15 U_1 \delta_{\mathrm{avg}}^2 - 18\delta_{\mathrm{avg}}^3) \right], \\
\mathcal{M}_{(8,6)}^h &= \left[ 4U_1(U_1 - 9\delta_{\mathrm{avg}}) \right], \\
\mathcal{M}_{(10,4)}^h &= \left[ U_1(U_1 - 3\delta_{\mathrm{avg}})(3U_1^4 - 18U_1^3 \delta_{\mathrm{avg}} + 55U_1^2 \delta_{\mathrm{avg}}^2 - 84U_1 \delta_{\mathrm{avg}}^3 + 85\delta_{\mathrm{avg}}^4) \right], \\
\mathcal{M}_{(10,6)}^h &= \left[ 2U_1(282U_1^3 - 631U_1^2 \delta_{\mathrm{avg}} + 126U_1 \delta_{\mathrm{avg}}^2 + 615\delta_{\mathrm{avg}}^3) \right], \\
\mathcal{M}_{(10,8)}^h &= \left[ U_1(337U_1 + 975\delta_{\mathrm{avg}}) \right],
\end{aligned}
$$

and one from the mutual effect of Hamiltonian at different times is

$$C_{R=1,\mathrm{triangle}}^{\mathrm{Time}}(T) = -\frac{T^8 \Omega^4}{20736} \left[ 1 + \frac{T^2}{288}(\delta_f - \delta_0)^2 \right] (\delta_f - \delta_0)^2 \mathcal{M}_{(6,4)}^h. \tag{A.33}$$

Similarly, $C_{R=1,\mathrm{triangle}}^{\mathrm{Path}}$ and $C_{R=1,\mathrm{triangle}}^{\mathrm{Time}}$ are respectively identical with $C^{\mathrm{Path}}$ and $C^{\mathrm{Time}}$ in the nearest-neighbor correlations of square lattice due to the same shortest path.

To verify the validity of ME for more complex structures, we examine the nearest-neighbor AF correlation in the triangular lattice geometry, a prototype of frustrated magnetic systems [see Fig. 12 (a)]. Frustrated magnetic phenomena are macroscopic manifestations of many-body physics, where the interplay of degenerate quantum states leads to macroscopic AF correlations. Frustrated magnets can exhibit exotic phases of matter, such as spin liquids, in which spins are disordered but remain in a liquid-like state due to strong quantum fluctuations, even at absolute zero temperature. Spin liquids are promising candidates for quantum communication and computation due to their unique properties, including long-range entanglement and fractional quantum excitations [36].

Therefore, the study of frustrated magnets is critical for deepening our understanding of fundamental physics and holds significant potential for the development of advanced materials and technologies. From our analytical calculations, we obtain identical expressions for

(a)

(b)

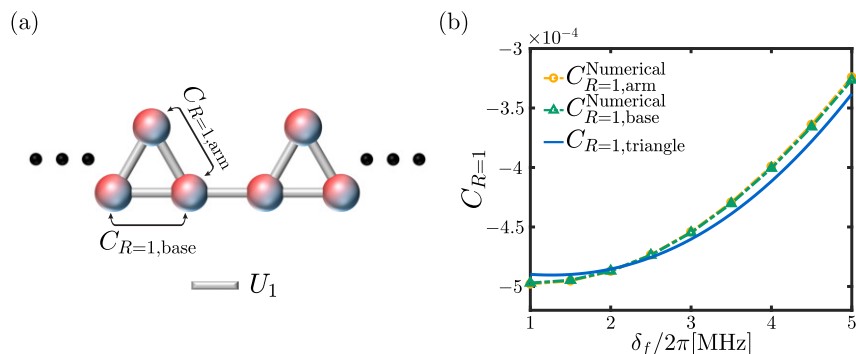

Figure 12: The buildup of antiferromagnetic correlation on triangular lattice. (a) Schematic descriptions of this lattice. (b) The nearest-neighbor correlations $C_{R=1}$ as the function of $\delta_f$. The yellow dotted line with circle and the green dotted line with triangle are produced by the numerical results. The blue solid line shows the analytic result of the corresponding correlation functions on the local lattice geometry.

the nearest-neighbor correlation between any two atoms in the triangular lattice. Specifically, the terms from the shortest path and the mutual effects of the Hamiltonian at different times are consistent with those in the $2 \times n$ lattice. That is, $C_{R=1,\text{triangle}}^{\text{Path}} = C_{10,\text{square}}^{\text{Path}}$, and $C_{R=1,\text{triangle}}^{\text{Time}} = C_{10,\text{square}}^{\text{Time}}$. Fig. 12 (b) shows that the ME results are in good agreement with the numerical calculations. Despite the complex evolution of degenerate quantum in the frustrated magnetic system, the ME still provides a quantitative description of the buildup of the AF correlation. Furthermore, $C_{R=1,\text{arm}}$ is approximately identical to $C_{R=1,\text{base}}$, indicating that the nearest-neighbor sites coupled to the triangle sites have a negligible effect on the buildup of AF correlation.

# B  The self-consistency of Magnus expansion

As a controlled approximation scheme, the ME must satisfy fundamental physical constraints at every truncation order and in the full resummation. A critical benchmark requires that when interactions vanish ($U = 0$), all connected spin-spin correlations must strictly vanish ($C_R(T) \equiv 0$). We analytically prove that within the ME framework, each individual expansion order independently enforces $C_R(T) = 0$ at $U = 0$, thereby preserving this fundamental causality condition. Formally, the connected correlation function between the $(0,0)$ and $(ka, lb)$ sites is defined as:

$$C_R(T) = \mathcal{L} - L, \tag{B.1}$$

where $\mathcal{L} = \langle \psi(T)|n_{(0,0)} n_{(k,l)}|\psi(T)\rangle$, and $L = \langle \psi(T)|n_{(0,0)}|\psi(T)\rangle \langle \psi(T)|n_{(k,l)}|\psi(T)\rangle$.

We begin by considering only the $\Omega$-term in a two-site system, where the Hamiltonian is $H = \frac{\Omega}{2}\sigma_1^x + \frac{\Omega}{2}\sigma_2^x$. The term $\Omega$ flips the spin state, such that the excited state $|r\rangle$ occurs only when $\Omega^n$ with $n \in odd$ acts on the ground state $|g\rangle$. Using this, we derive $\mathcal{L}$ as follows:

$$\mathcal{L} = \sum_{m,n\in even} \sum_{l,l'} \frac{(iT)^m}{(2l+1)!(m-2l-1)!} \frac{(-iT)^n}{(2l'+1)!(n-2l'-1)!} \left(\frac{\Omega}{2}\right)^{m+n}$$
$$\times \left\langle \left(\sigma_1^x\right)^{2l+1} \left(\sigma_2^x\right)^{m-2l-1} \left(\sigma_1^x\right)^{2l'+1} \left(\sigma_2^x\right)^{m-2l'-1} \right\rangle,$$

where $\sigma_1^x$ and $\sigma_2^x$ contributes identical magnitudes, which are independent of the site position.

So we can simplify $\mathcal{L}$ into

$$\mathcal{L} = \sum_{m,n=2}^{\infty} \frac{(iT)^m 2^{m-1}}{m!} \frac{(-iT)^n 2^{n-1}}{n!} \left\langle \left(\frac{\Omega}{2}\sigma^x\right)^{m+n} \right\rangle$$

$$= \sum_{m,n=2}^{\infty} \frac{(iT)^m}{m!} \frac{(-iT)^n}{n!} \frac{\left\langle (\Omega\sigma^x)^{m+n} \right\rangle}{4}.$$

Similarly, $L$ is simplified as,

$$L = \sum_{p,q,k,l=1}^{\infty} \frac{(iT)^p}{p!} \frac{(-iT)^q}{q!} \frac{(iT)^k}{k!} \frac{(-iT)^l}{l!} \frac{\left\langle (\Omega\sigma^x)^{p+q+k+l} \right\rangle}{16},$$

where $p, q, k, l \geq 1$ and $p + q + k + l = m + n = N$.

To confirm $\mathcal{L} = L$, we need to prove:

$$\sum_{m,n=2}^{\infty} \frac{4}{m!n!} = \sum_{p,q,k,l=1}^{\infty} \frac{(-1)^{q+l}}{p!q!k!l!}. \tag{B.2}$$

Expanding both sides, we find the left-hand side,

$$\sum_{m,n=2}^{\infty} \frac{4}{m!n!} = \frac{1}{N!} \sum_{m,n=2}^{\infty} \frac{4N!}{m!(N-m)!} = \frac{1}{N!}(2^{N+1} - 8),$$

and the right-hand side,

$$\sum_{p,q,k,l=1}^{\infty} \frac{(-1)^{q+l}}{p!q!k!l!} = \sum_{m',n_e} \frac{1}{m'!n_e!}(2^{m'} - 2)(2^{n_e} - 2) - \sum_{m',n_o} \frac{1}{m'!n_o!}(2^{m'} - 2)(2^{n_o} - 2)$$

$$= \sum_{n_e} \frac{1}{(N-n_e)!n_e!}(2^{N-n_e} - 2)(2^{n_e} - 2) - \sum_{n_o} \frac{1}{(N-n_o)!n_o!}(2^{N-n_o} - 2)(2^{n_o} - 2)$$

$$= \frac{1}{N!}(2^{N+1} - 8),$$

where $m' = p + k, n' = q + l, m' + n' = N$, $n_e \in even, n_o \in odd$. Thus, $\mathcal{L} = L$ is verified for $U = 0$ at all expansion levels when only the $\Omega$-term is considered.

Next, we include the $\delta$-term in the Hamiltonian, $H = \frac{\Omega}{2}\sigma_1^x + \frac{\Omega}{2}\sigma_2^x - \delta n_1 - \delta n_2$. Since $[\sigma_x, n] \neq 0$, $H^m$ cannot be decomposed into grid-independent terms. However, for any superposition state $|\psi\rangle = a|g\rangle + b|r\rangle$ where $|a|^2 + |b|^2 = 1$, we find $\langle \psi | [\sigma_x, n] | \psi \rangle = 0$. Hence, the Hamiltonian can be expressed as:

$$H^m = \left(\frac{\Omega}{2}\sigma_1^x + \frac{\Omega}{2}\sigma_2^x - \delta n_1 - \delta n_2\right)^m = \sum_{l=1}^{\infty} C_m^l \left(\frac{\Omega}{2}\sigma_1^x - \delta n_1\right)^l \left(\frac{\Omega}{2}\sigma_2^x - \delta n_2\right)^{m-l}$$

$$= \sum_{l=1}^{\infty} \frac{m!}{l!(m-l)!} \left(\frac{\Omega}{2}\sigma^x - \delta n\right)^m.$$

The corresponding two components of the correlation function can be written as:

$$\mathcal{L} = \sum_{m,n} \frac{(iT)^m}{m!} \frac{(-iT)^n}{n!} (2^m - 2)(2^n - 2)\left\langle \left(\frac{\Omega}{2}\sigma^x - \delta n\right)^N \right\rangle,$$

$$L = \sum_{p,q,k,l} \frac{(iT)^p}{p!} \frac{(-iT)^q}{q!} \frac{(iT)^k}{k!} \frac{(-iT)^l}{l!} (2^p - 2)(2^q - 2)(2^k - 2)(2^l - 2)\left\langle \left(\frac{\Omega}{2}\sigma^x - \delta n\right)^N \right\rangle.$$

Using a similar approach as in Eq. (B.2), the terms in $\mathcal{L}$ and $L$ can be expanded as:

$$\mathcal{L} = \sum_{m,n} \frac{1}{m!} \frac{1}{n!} (2^m - 2)(2^n - 2) = \frac{1}{N!} \left( 2^{N+1} - 8 \right),$$

and

$$
\begin{aligned}
L &= \sum_{p,q,k,l} \frac{1}{p!} \frac{1}{q!} \frac{1}{k!} \frac{1}{l!} (2^p - 2)(2^q - 2)\left( 2^k - 2 \right)\left( 2^l - 2 \right) \\
&= \sum_{m',n'} \sum_{p,q} \frac{1}{p!(m'-p)!} \frac{(-1)^{n'}}{q!(n'-q)!} \left( 2^{m'} - 2^p - 2^{m'-p} + 1 \right)\left( 2^{n'} - 2^q - 2^{n'-q} + 1 \right) \\
&= \sum_{m',n'} (-1)^{n'} \frac{4^{m'} - 2 \times 3^{m'} + 2^{m'}}{m'!} \frac{4^{n'} - 2 \times 3^{n'} + 2^{n'}}{n'!} \\
&= \frac{1}{N!} (2^{N+1} - 8).
\end{aligned}
$$

Thus, we conclude that $\mathcal{L} = L$ for $U = 0$, indicating the $C_R(T)$ vanishes in every perturbative order due to the absence of interaction.

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
