# Peer review of "Algebraic law of local correlations in a driven Rydberg atomic system"

_SciPost Physics, doi:SciPost Phys. 19, 152 (2025)_

## Round 1 · Referee Report · Anonymous (Referee 1) · 2025-9-12

Strengths

1 - very detailed results

Weaknesses

1 - Magnus expansion validity unclear

Report

The authors show that certain long-range interacting models consistent with Rydberg systems have algebraic law of correlations. This seems to be related to the long-range nature of the models. Moreover, the authors imply that this result is universal.

Requested changes

I propose the following changes:

1 - I would not say that exponential growth of HS renders mean field intractable (31) 2 - The main result should be described better in the Abstract: "The AF correlation magnitude at fixed Manhattan distance emerges as the algebraic sum of contributions from all shortest paths, regardless of their topological equivalence." (49 etc). It is not clear what the main result is 3- Fix typo "Magnuns"(139) 4- I am not convinced that their Magnus expansion is completely correct, but I do not object to its use especially since it is consistent with numerics. I would suggest that they remark on this issue.

Recommendation

Ask for major revision

  • validity: ok
  • significance: good
  • originality: high
  • clarity: ok
  • formatting: good
  • grammar: good

Author:  Bo Xiong  on 2025-09-15  [id 5817]

(in reply to Report 1 on 2025-09-12)
Category:
answer to question

We thank the Referee’s thoughtful comments. We have revised the manuscript accordingly and provide our responses below:

1 - After reconsidering our original statement, we agree that it should be clarified. We prefer to say that the inhomogeneous interactions render the mean-field expansion intractable. Thus we have revised it to more accurately reflect the respective roles of these factors: “The exponential growth of the Hilbert space prevents exact solutions [7] and challenges numerical simulations, while the spatially inhomogeneous interactions render mean-field expansions [8] intractable.”

2 - We have revised the sentence (49 etc): “The AF correlation magnitude at fixed Manhattan distance is governed by the algebraic sum of contributions from all shortest paths. This law is independent of path equivalence, lattice geometries, and quench protocols, implying that complex correlation networks can be deconstructed into elementary path contributions.”

3 - The word “Magnuns” has been corrected to “Magnus”.

4 - We have added the statement about the validity of ME (159 etc): “Although the Magnus expansion is an approximate method, the quantitative agreement between our analytical results and numerical simulations in various geometries and quench protocols supports the validity of the second-order ME for capturing the essential physics of the AF correlation buildup under the parameters and time scales considered. Furthermore, the rigorous proof of the self-consistency of ME, e.g., for $U=0$, $C_{kl}=0$, is shown in Appendix B.”

---

## Round 1 · Referee Report · Anonymous (Referee 2) · 2025-10-7

Strengths

1) The paper develops a method to study the bulid up of antiferromagnetic correlations in a Rydberg atom platform.

2) The results are detailed.

Weaknesses

  1. Use of Magnus expansion.

Report

The authors study buld up of antiferromagnetic spin model which is realizable in Rydberg platforms using a Magnus expansion. I think that the paper requires serious modification and here ar ethe relevant points.

a) The use of Magnus expanion means that the results can only be shown for
high frequency ( even when we discuss finite time). The authors needs to discuss this more clearly. In particular, they need to point out the radius of convergence of this expansion in the regime they are interested in.

b) The authors show gradual build up of the correlations order by prder in perturbatyion theory. This might lead the authors to think that this build up is a property of the Magnus expansion. The authors, to counter this point, could, for example use Floquet Perturbation theory (or Magnus in a rotated frame)
which has a wider valkidity range. I thnk this point needs to be chenked to ascertain the universality of the results.

In absence of clarification of these issues, I can not recommed Scipost Phys. The paper is certainly publsihable in its present version, but possibly in Scipost Core.

Requested changes

1) Check for FPT ( see report) 2) Comment on domain of valifdity of the Mgnus expansion ( see report).

Recommendation

Accept in alternative Journal (see Report)

  • validity: ok
  • significance: ok
  • originality: high
  • clarity: good
  • formatting: good
  • grammar: good

Author:  Bo Xiong  on 2025-10-27  [id 5953]

(in reply to Report 2 on 2025-10-07)

1) We thank the referee for this insightful suggestion, which prompted us to further substantiate the universality of our central finding -- the path superposition principle. We agree that relying solely on the convergence of the Magnus expansion (ME) could raise concerns about its method-dependence. Our primary evidence for universality stems from the \textbf{consistent quantitative agreement} between our second-order ME results and \textbf{exact numerical simulations} across various systems, including different lattice geometries and quench protocols. This robust agreement strongly suggests that the superposition law captures the underlying physics of correlation buildup during the quench dynamics, rather than being an artifact of the perturbative method. Regarding the suggestion to employ Floquet Perturbation Theory (FPT), we note that our work focuses on \textbf{single-quench dynamics}, for which the ME is a natural and suitable analytical tool. FPT, while powerful for periodically driven systems, is formally designed for Floquet steady states and may not be directly applicable or necessary for our transient quench scenario. The excellent performance of ME in our context justifies its use. Nevertheless, exploring whether the superposition law extends to periodically driven systems using FPT is a compelling future direction. We have therefore added a discussion in the conclusion: “{\it Investigating the persistence of the path superposition principle in periodically driven systems, for instance using Floquet Perturbation Theory [New J. Phys. 20, 093022 (2018)], represents an important future research direction.}”

2) We thank the referee for raising this crucial point regarding the domain of validity of the Magnus expansion (ME). Our application of the ME indeed differs from its conventional use, and we appreciate the opportunity to clarify this distinction. The referee is correct that when the ME is used to derive a static \textbf{Floquet Hamiltonian} for a periodically driven system, its convergence requires a high-frequency condition ($\Omega \gg $ energy scales) to accurately capture the long-time stroboscopic dynamics. However, our work employs the ME in a fundamentally different context. We study a \textbf{single, non-periodic quench} rather than a periodic drive. Here, the ME is not used to find a Floquet Hamiltonian but to construct a \textbf{finite-time evolution operator} $\hat{U}(T)$ directly. The convergence criterion for this application is a \textbf{short-time condition}, not a high-frequency one. The standard sufficient condition for the convergence of the ME series is $\int_{0}^{T}\Vert H(t)\Vert_{2}\, dt < \pi$, which, for our quench protocol, implies an upper bound on the product of the quench amplitude and duration, $\vert \Delta \delta \vert \cdot T$. Therefore, the “radius of convergence” in our context is defined by the \textbf{finite quench time $T$ being sufficiently short}. We provide two key evidences that our chosen duration ($T = 0.5 \mu s$) lies well within this convergent regime: (i) \textbf{Empirical Numerical Agreement:} As shown throughout the manuscript, our second-order ME results are in quantitative agreement with exact numerical simulations across various lattice geometries. (ii) \textbf{Theoretical Self-Consistency:} As rigorously shown in Appendix B, our expansion correctly yields null correlations for the non-interacting case ($U = 0$) at any truncation order, satisfying a fundamental physical constraint and validating the internal consistency of our formalism. For $U \neq 0$, we have systematically verified the validity of our approach. We find that quantitative agreement with exact numerics (see Fig.\,8 in Appendix A) is achieved only when our expansion incorporates all key physical processes: (i) the shortest path; (ii) couplings from neighboring sites; (iii) temporal interference from interaction time-ordering. In summary, the high-frequency limit is not pertinent to our single-quench problem. The ME, governed by a short-time convergence criterion, is exceptionally well-suited for providing an accurate and analytically transparent description of our quench dynamics, successfully revealing the universal path superposition law.

Attachment:

2ndReply.pdf

---

## Round 2 · Referee Report · Anonymous (Referee 2) · 2025-10-27

Report

The authors have taken care of all the issues raised in the previous report. In particular their analysis of the radius of convergence of the ME is satisfactory; the validity of their results is argued in a convincing manner. In view of this and the fact that the paper addresses an important issue in an experimentally realizable ultracold atom platform, I think I can recommend publication in Scipost Phys.

Recommendation

Publish (easily meets expectations and criteria for this Journal; among top 50%)

---

## Round 2 · Referee Report · Anonymous (Referee 1) · 2025-11-5

Report

The authors have addressed the concerns raised, mainly the Magnus expansion. I agree with the authors that their work opens a new clear multipronged approach for future work - in particular their quite general result of a universal superposition principle of an algebraic sum of contributions from shortest paths. Hence, I now strongly recommend publication.

Recommendation

Publish (easily meets expectations and criteria for this Journal; among top 50%)

---

## Editorial Decision

published